

# Best practices for estimating turbulent dissipation from oceanic single-point velocity timeseries observations

Cynthia E. Bluteau[1], Danielle Wain[2], Julia C. Mullarney[3], and Craig L. Stevens[4,5]

[1]Institute of Ocean Sciences, Fisheries and Oceans Canada, Sidney BC, Canada
[2]7 Lakes Alliance, Belgrade Lakes ME, USA
[3]Coastal Marine Group, School of Science, University of Waikato, Hamilton, New Zealand
[4]Institute for Earth Science, Wellington, New Zealand
[5]Dept. Physics, University of Auckland, Auckland, New Zealand

**Correspondence:** Cynthia E. Bluteau (cynthia.bluteau@dfo-mpo.gc.ca)

**Abstract.** We provide best practices for estimating the dissipation rate of turbulent kinetic energy, $\varepsilon$, from velocity measurements in an oceanographic context. These recommendations were developed as part of the Scientific Committee on Oceanographic Research (SCOR) Working Group #160 "Analyzing ocean turbulence observations to quantify mixing". The recommendations here focus on velocity measurements that enable fitting the inertial subrange of wavenumber velocity spectra.

The method examines the measurable range for this method of dissipation rates in the ocean, seas, and other natural waters. The recommendations are intended to be platform-independent since the velocities may be measured using bottom-mounted platforms, platforms mounted beneath the ice, or platforms directly on mooring lines once the data is motion-decontaminated. The procedure for preparing the data for spectral estimation is discussed in detail, along with the quality control metrics that should accompany each estimate of $\varepsilon$ during data archiving. The methods are applied to four 'benchmark' datasets covering

different flow regimes and two instrument types (acoustic-Doppler and time of travel). Problems associated with velocity data quality, such as phase-wrapping, spikes, measurement noise, and frame interference, are illustrated with examples drawn from the benchmarks. Difficulties in resolving and identifying the inertial subrange are also discussed, and recommendations on how these issues should be identified and flagged during data archiving are provided.



## 1 Introduction

Quantifying turbulence and mixing in the ocean is critical for understanding many processes including the turbulent transfer of momentum, heat and material —as well as the dissipation of energy (Fox-Kemper et al., 2019). For example, when seeking to understand how an ocean boundary layer responds to wind, it is critical to reliably understand how energy is dissipated by a turbulent cascade to the viscous scales —or how material is exchanged across isolines, such as the seasonal thermocline, as well as through boundaries of the fluid into the benthos or atmosphere (D'Asaro, 2014).

However, owing to the small temporal and spatial scales which inherently characterize turbulent flow, key representative quantities are often not straightforward to measure (Gregg, 1999). The dissipation rate of turbulent kinetic energy $\varepsilon$ describes the energy lost from the fluid system to viscous dissipation. In addition, $\varepsilon$ can be related to processes that drive turbulent fluxes of momentum, heat and material. The small scales and variable nature of the processes require precise measurement and analysis before one arrives at an estimate of $\varepsilon$ and associated quantities.

These computations are especially challenging in the ocean as $\varepsilon$ can range over almost ten orders of magnitude (Stewart and Grant, 1999). Consequently, distinct measurement techniques have been developed for differing parts of this wide range of $\varepsilon$. Microstructure profiling shear probes can measure turbulence in low to medium energy regions of the ocean (Lueck et al., 2024), while typical point-velocity sensors allow for estimating timeseries of dissipation rate in higher energy environments than shear probes.

Acoustic point-measurement instruments —sensors which measure point-velocity timeseries in the ocean —have been deployed since the mid-1990s (Lemmin et al., 1999; Rusello et al., 2006), and are based on either acoustic backscatter (Trowbridge and Elgar, 2001), or time of travel (Henchicks, 2001) approaches. The time of travel instruments (e.g., MAVS) are better suited to low-energy environments than the acoustic backscatter approach. The single-point measurements were found to be especially well-suited to capture $\varepsilon$ values within boundary-layers to improve understanding of dynamics and transfer processes
(Li et al., 2022); or, in experiments in which the control volume is fixed in space - i.e., the sensor is mounted on the sea bed or a fixed platform, rather than in open-ocean measurements of $\varepsilon$ such as recorded with a falling profiler (Le Boyer et al., 2023).

Technology for acoustic point velocity timeseries measurements has matured over the last decades, with many commercial instruments now well-established (e.g., the Nortek Acoustic Doppler Velocimeter, ADV, and Nobska Modular Acoustic Velocity Sensor, MAVS), resulting in a wide user base of researchers and engineers. In particular, improvements in noise reduction
meant that by the late 1990s, data from these instruments were able to be analysed to provide turbulence estimates (e.g., Voulgaris and Trowbridge, 1998; Kim et al., 2000). Previously, such estimates were only undertaken by a small number of groups worldwide with bespoke equipment that was built, maintained, and operated in-house. With this expansion in the user-base comes a need for consistent, or at least comparable, methods to assess data quality and archive datasets. However, currently no standardized methods or guidelines exist for these velocity measurements.

Here, we address this deficit by describing recommendations for best practices for obtaining $\varepsilon$ from point velocity measurements. These recommendations were developed through the Scientific Committee on Oceanic Research working group #160 on Analysing ocean Turbulence Observations to quantify MIXing (ATOMIX), specifically the subgroup on point velocities.



The working group established benchmark datasets to assess and validate algorithms (independent of programming language)
for estimating $\varepsilon$.

In addition, ATOMIX developed recommendations for two other types of technologies, shear probes (Lueck et al., 2024),
and acoustic-Doppler current profilers (in prep). The ATOMIX approach promotes a consistent variable naming framework,
followed by example-based description of guidelines for formatting, processing level, parameter estimation, and quality control.
A key result is the provision of final $\varepsilon$ estimates for the benchmark datasets to enable researchers to check their own analyses.

## 2    Background theory and sampling requirements

A critical aspect of most approaches to sampling environmental turbulence is that data are typically acquired in the time
domain, but the mechanistic, physical understanding is best described and analyzed in the spatial (i.e., wavenumber) domain.
Technological developments are on the cusp of enabling the direct measurement of spatial spectra (e.g., Shcherbina et al.,
2018), but we focus on the more traditional timeseries measurements. These measurements are converted from a time to spatial
frame of reference by invoking Taylor's frozen turbulence hypothesis whereby, under certain conditions, turbulent structure
can be considered 'frozen' as it moves past a stationary sensor (Lumley, 1965; Wyngaard and Clifford, 1977). In practice,
this hypothesis requires an estimate of the mean advection speed $\bar{U}$ past the sensor, which enables the conversion of spectral
observations $\hat{\Psi}$ from the frequency $f$ (Hz) to wavenumber domain $\tilde{k}$ (cpm) as:

$$\hat{\Psi}(\tilde{k}) = \bar{U}\hat{\Psi}(f)$$
$$\tilde{k} = \frac{f}{\bar{U}}. \tag{1}$$

The presence of $\hat{\cdot}$ above variables indicate observational or estimated parameters.

Irrespective of whether the instrument is fixed or moving (e.g., drifting or moored), the velocity $\bar{U}$ is always the speed
relative to the instrument rather than the actual water speed. This conversion can lead to errors in $\varepsilon$ of a few percent when
$\bar{U}$ is about ten times larger than the root-mean-square of the turbulent velocity fluctuations in the direction of the mean flow
(Wyngaard and Clifford, 1977; Lumley, 1965). The error magnitude in flows with low mean speeds $\bar{U}$ was determined more
recently by Pécseli and Trulsen (2022), using idealized numerical experiments. Their work excludes the impact of surface
waves on the advection of turbulence and the estimated $u_{\mathrm{rms}}$. They found that the expected errors on $\hat{\varepsilon}$ drops when the ratio
$u_{\mathrm{rms}}/\bar{U}$ increases. When $\bar{U}$ was three times larger than $u_{\mathrm{rms}}$, the $\hat{\varepsilon}$ estimates were overpredicted by less than 10%. The errors
grew to about 25% when $\bar{U}$ was comparable in magnitude to $u_{\mathrm{rms}}$ (Pécseli and Trulsen, 2022).

Taylor's hypothesis impacts which wavenumbers are resolved in the final spectra. Slower mean speeds enable resolution of
the smaller turbulence scales within the viscous subrange without increasing the sampling rate of measurements (Figure 2).
Hence, the expected mean speed $\bar{U}$ must be considered when selecting the sampling frequency of measurements and setting
the ambiguity velocity of the instrument. A low ambiguity velocity in pulse-coherent Doppler instruments improves the data
quality by reducing the measurement noise but might "phase wrap" the measured velocities (Lhermitte and Serafin, 1984;





**Figure 1.** Sketch of point velocity deployment configurations showing both under ice and seafloor deployments with examples of a time of travel sensor (upper) or acoustic backscatter sensor (lower).

Lohrmann and Nylund, 2008). Section §4.1 covers how phase-wrapped data can be identified and remedied, while the metrics for assessing the validity of Taylor's hypothesis will be discussed in section 4.2.

**Figure 2.** Spectral representations from the four benchmark datasets overlaying the expected idealized curves for a range of $\varepsilon$. The gray diamonds denote the empirical limit of the inertial subrange and depend on $\varepsilon$. The coloured triangles represent the approximate distance of the platform from the nearest boundary ($2\pi/\kappa z$). The impact of vortex shedding is apparent in the under-ice MAVS example at approximately 25 cpm.

Estimating turbulence quantities from field measurements also necessitates satisfying the stationarity assumption. This assumption implies that the statistical properties of the flow do not vary with time; and hence, $\varepsilon$ does not evolve faster than the timescales over which the value is estimated. Stationarity must be considered when choosing the time over which to estimate





$\varepsilon$ (see § 4.2). More importantly, however, the sampling rate and segmenting choices must consider the time and length scales associated with ocean turbulence, particularly those corresponding to the inertial subrange (Figure 2).

The inertial subrange spectral model $\Psi_j(k)$ is given by:

$$\Psi_j(k) = a_j C_k \varepsilon^{2/3} k^{-5/3}, \tag{2}$$

where $k$ is the wavenumber expressed in rad m$^{-1}$ rather than $\tilde{k}$ expressed in cpm ($\tilde{k} = 2\pi k$), $C_k$ is the empirical Kolmogorov universal constant that is approximately $C_k = 1.5$ as given in Sreenivasan (1995). The constant $a_j$ depends on the velocity component for estimating $\varepsilon$. In the longitudinal direction $a_1 = 18/55$, while the values are 4/3 larger in the vertical and trans-

verse directions, i.e. $a_2 = a_3 = 4a_1/3$ (Pope, 2000). Sreenivasan (1995) collated and interpreted multiple studies to obtain an average value for $C_k$=1.62 ($a_1 C_k = 0.53$) with a standard deviation of 0.1681 (or 0.055 for $a_1 C_k$).

The inertial subrange covers larger scales than the viscous subrange. It can be nonexistent in low Reynolds number flow in which the larger turbulent scales become comparable in size to the smallest scales (Saddoughi and Veeravalli, 1994; Gargett et al., 1984). The largest scales depend on the sought-after quantities – $\varepsilon$ and the background flow properties. The smallest

scales are defined by the Kolmogorov length scale $L_K$:

$$L_K = \left(\nu^3/\varepsilon\right)^{1/4}, \tag{3}$$

where $\nu$ is the kinematic viscosity of seawater. The largest scales of the inertial subrange are about ten times the Kolmogorov scale, which translates to:

$$k_{is} \approx \frac{1}{10L_K} \tag{4}$$

(Pope, 2000). When the ratio between the largest and smallest turbulent length scale becomes smaller than roughly 300, the inertial subrange becomes severely anisotropic (see the review in Bluteau et al., 2011). Energy levels drop below those predicted by the isotropic model in equation 2, and the inertial subrange becomes unsuitable for estimating $\varepsilon$. These problems typically occur for weakly turbulent flows, especially near boundaries or in highly stratified-sheared flows (Gargett et al., 1984; Saddoughi and Veeravalli, 1994).

Several relationships exist to define the scale of the largest turbulent overturns (Ivey et al., 2018). We provide a brief overview, noting that these scales can be of the order of meters. The scales are always limited by the distance to the nearest boundary, either the bottom or the surface. One common definition for the large overturns in a stratified-sheared flow is the Ozmidov length scale $L_O$ (Ozmidov, 1965):

$$L_O = \left(\varepsilon/N^3\right)^{1/2}, \tag{5}$$

or by the Corssin length scales in sheared flows (Corrsin, 1958):

$$L_S = \left(\varepsilon/S^3\right)^{1/2}, \tag{6}$$

where $N$ and $S$ are the background stratification and velocity shear. This length scale tends to be smaller than $L_O$ and better represents the low wavenumber limit of the inertial subrange (see, for example Bluteau et al., 2011).





Near boundaries, other definitions for the largest overturn size may be warranted. For example, the Obukhov length scale for
ocean convection includes the effects of buoyancy and applied wind stress (Obukhov, 1946; Zheng et al., 2021). Near a solid
boundary,

$$L_S \equiv \kappa z \tag{7}$$

when the assumptions of the log-law of the wall are satisfied (Bluteau et al., 2011). The Von Kàrmàn's constant is $\kappa = 0.39$,
as revised by Marusic et al. (2013) using atmospheric and laboratory observations over a wide range of Reynolds numbers.
Hence, when no velocity shear measurements are available, the distance of the nearest boundary can be used to characterize
the largest overturns, although this approach may overestimate the overturn sizes.

Generally, the inertial subrange can be identified directly from the spectral observations. Knowledge of the above length
scales is important to ensure the sampling and analysis strategies do not inadvertently reduce the range of wavenumbers
resolved within the inertial subrange. These scales partly dictate the segmenting and spectral averaging strategies described
below (§ 4.2 and 4.3), as well as the choice of burst sampling duration if continuous sampling is unfeasible with the available
battery power of the instrument.

Measurement campaigns must ensure the sampling rate is sufficiently fast to resolve the high, most isotropic, wavenumbers
of the inertial subrange. The sampling rate must be faster for fast-moving flows than in low-energy environments, although
the high-energy flows typically lead to a wider inertial subrange. A sampling rate of 64 Hz has a Nyquist frequency of 32 Hz.
If the noise levels are low, the highest wavenumbers of the inertial subrange are resolved for $\varepsilon \lesssim 10^{-5}\,\mathrm{W\,kg^{-1}}$ and mean
speeds $\bar{U} \lesssim 1\,\mathrm{m\ s^{-1}}$. For slower expected speeds $\bar{U} \lesssim 0.25\,\mathrm{m\,s^{-1}}$, a sampling rate of about 16 Hz suffices for resolving the
entire inertial subrange when $\varepsilon \lesssim 10^{-5}\,\mathrm{W\,kg^{-1}}$. The sampling rate can be further reduced if the expected $\varepsilon$ is much lower than
$10^{-5}\,\mathrm{W\,kg^{-1}}$ (see Figure 2). However, the noise floor adversely impacts our ability to estimate low $\varepsilon$ by potentially drowning
out the high wavenumbers of the inertial subrange (§4.4.3).

# 3 Benchmark datasets and formatting

The benchmark datasets were selected to cover a range of instrument types and environmental conditions that can be encoun-
tered ranging from low-energy environments, such as beneath sea ice or lakes, to high-energy environments, such as sills and
obstacles in coastal oceans or shallow embayments and estuaries (Bluteau et al., 2025). Here, we focus on four benchmark
datasets encompassing a range of water depths and background flow speeds (Table 1). Other datasets were considered, but the
present ones have problematic sections that allowed us to demonstrate the application of quality control metrics.

All four datasets are characterized by being recorded relatively close to a horizontal boundary. The four benchmark datasets
are split evenly across two types of instruments - (i) acoustic-Doppler velocimeter (ADV) which is a backscatter device, and
(ii) Modular Acoustic Velocity Sensor (MAVS) which is a time-of-travel device. The ADV datasets presented here were both
collected with a Vector, Nortek AS, while the MAVS instruments were produced by Nobska. The MAVS instrument notably
does not require particulate material in the water column to resolve speed fluctuations and, thus, turbulence. However, as we




discuss later, their sampling rings shed vortexes that contaminate velocities in the direction perpendicular to the instrument's main shaft (Hay et al., 2013). Bottom frames may also contaminate MAVS and ADV velocity measurements.

We specify the technology type used for collecting the velocities in the names of the benchmark datasets, which can be summarized as follows:

1. *Tidal Slough ADV* - unstratified, shallow water;

2. *Tidal Shelf ADV* - stratified boundary-layer with relatively fast speeds, in deep water;

3. *Under-ice MAVS* - weak flows, 5 m beneath rough ice;

4. *Tidal MAVS* - fast flows, near bed in shallow and unstratified waters.

**Table 1.** Summary of setup, environmental conditions, and estimated $\hat{\varepsilon}$ for benchmark datasets (Bluteau et al., 2025). The range for mean speed $\bar{U}$ represents the $50^{\text{th}}$ and $95^{\text{th}}$ percentile, while the range for $\hat{\varepsilon}$ represents the $2.5^{\text{th}}$ to $97.5^{\text{th}}$ percentile.

| Name | Water depth | $z$ | $\bar{U}$ | $\tau_\varepsilon$ ($\tau_{\text{FFT}}$) | sampling rate | $\hat{\varepsilon}$ | Comments |
|---|---|---|---|---|---|---|---|
| | m | m | $\text{m s}^{-1}$ | seconds | Hz | $\text{W kg}^{-1}$ | |
| Tidal slough ADV | 2.8 | 0.45 | 0.15-0.33 | 540 (135) | 16 | 1e-7 to 1e-5 | Tidal slough deployment where the viscous subrange is occasionally resolved. Unstratified, but shear-induced anisotropy |
| Tidal shelf ADV | 250 | 0.4 | 0.25-0.78 | 256 (32) | 64 | 1e-7 to 1e-4 | Continental shelf deployment in a Stratified bottom log layer. Dataset has phase wrapping and flow-dependent evidence of vortex shedding. |
| Under-ice MAVS | 353 | 5 | 0.03-0.06 | 1024 (256) | 8 | 3e-9 to 7e-8 | Slow under-ice and weakly-stratified, boundary layer in deep water. The instrument was suspended beneath crystal-coated ice. |
| Tidal MAVS | 20 | 1.45 | 1.0-1.1 | 82.8 (20.7) | 12.35 | 3e-4 to 1e-3 | Strong tidal flows in shallow water. Weak stratification |

The format of these benchmarks was devised to facilitate testing from different checkpoints, i.e., processing levels, and

include quality-control parameters to assess data quality. The benchmarks use the NetCDF-4 file format with four distinct processing levels stored in their own group named according to the processing levels (Figure 3). The variable dimensions were repeated within each group even though the last three processing levels shared many dimensions. This choice accommodates





the possibility of archiving only some processing levels in a central repository since archiving all levels may be prohibitive for long timeseries.

The first data level contains the raw velocity measurements and boolean quality-control indicators that flag poor-quality velocity samples. The second level involves applying the quality control flags, replacing the missing samples (typically through linear interpolation), and then segmenting the time series into smaller subsets, usually a few minutes long.

    This second level separates each subset of velocity samples used for computing spectra and other statistics required for converting from time to space or selecting turbulence models (e.g., wave statistics). The third level contains the spectral

observations for each segment that are used to derive $\varepsilon$ by fitting the observations over the inertial subrange with the appropriate theoretical model. The fourth level contains the estimated $\varepsilon$ from all available velocity components, along with boolean quality-control flags that indicate one or many reasons why an $\varepsilon$ should be discarded or, at the very least, have the quality questioned. The flags' metadata includes thresholds for any quality-control test applied to the original measurements. This level contains the $\varepsilon$, and quality-control indicators, which would typically be presented in a scientific article or technical report.

**4   Processing methods**

We detail the best practices for the processing step as they are applied to each NetCDF data level (Figure 3). Our processing choices were determined using the existing literature, ATOMIX members' experience, and testing of alternative methods. The current benchmarks and proposed best practices are a starting point for eventual processing standards for estimating $\varepsilon$. Our intention is for future users to verify their results at different data processing checkpoints, allowing the quality of archived

datasets to improve further over time.

**4.1   Quality control of raw velocity measurements**

A timeseries, recorded from one instrument, is stored as a two-dimensional matrix where each column represents a velocity component. The Level 1 flags are obtained by applying multiple quality-control processing steps to the raw measurements. We note that several of the quality control steps are applied to the velocities collected in beam coordinates. However, in acoustic

systems with three or four transducers, data is rotated into horizontal and vertical velocity components by applying linear transformations to the data in beam coordinates; thus, a 'bad data' flag in beam coordinates should generally also be applied in each of the xyz coordinates. Below, we describe identification of bad data and formation of the quality control flags as applied to the ADV benchmarks. The MAVS benchmarks have flags only at Level 4 for the $\hat{\varepsilon}$ estimates. The primary data quality issue for this instrument is vortex shedding from the sampling rings.

The most common indicators of low-quality data are either low backscatter amplitudes or low signal-to-noise ratios, which express the strength of the signal relative to a background noise level ($\text{SNR} = 20 \log 10(A_{\text{signal}}/A_{\text{noise}})$). Critical cut-off values for discarding data are instrument- and environment-specific. For the Nortek Vector used in the benchmark datasets, the manufacturer recommends values are 6 dB above the noise floor (around 50 dB) and 15 for amplitude and SNR, respectively (Nortek, 2018). In contrast, an obstruction of an acoustic beam may manifest as high values in both backscatter and SNR. This





**Figure 3.** Parallel data and NetCDF Level hierarchy showing the transition from raw time series to the final $\varepsilon$ timeseries with the segment analyses at each burst.

obstruction may be identified by a large difference in amplitude between the adjacent acoustic beams, especially if another beam is located behind the obstruction.

ADV systems use pulse-to-pulse coherent technology in which a pair of pulses separated by a known time lag are emitted. The similarity between the measured echo of the two pulses is assessed and reported as a percentage value, which provides a further indication of data quality (Lohrmann and Nylund, 2008). It is worth noting that coherence and amplitude are functions

of different parameters with varying sensitives, and thus provide two separate measures of quality control. Low correlation values can be used to identify bad data; but the converse is not necessarily true, that is, a large value doesn't indicate good quality data in all cases. A canonical cut-off value of 70% has been shown to generally reduce variance within the dataset; however, values of 50% are also commonly used as a cut-off. In general, values should be set on a case-by-case basis following careful inspection of the dataset (Nortek, 2018).





For pulse-coherent instruments, the Doppler phase shift can only be determined from $-\pi$ to $\pi$. For values outside this range, when the along-beam velocity exceeds the ambiguity velocity as set by the time-lag between pulses, 'phase wrapping' can occur. This effect manifests as a sudden and unrealistic change in velocities, which is usually also accompanied by a change in sign. These 'phase-wrapped' velocities can sometimes be corrected for in beam coordinates by subtracting or adding twice the ambiguity velocity. However, it is preferable to set the nominal maximum velocity to a sufficiently large value when

programming the instrument for deployment to reduce data processing issues (Rusello, 2009).

Further quality-control measures may involve removing excessively large speeds and/or other outliers, which could arise under some circumstances, such as broken instruments or biofouling. Our quality control data format provides users with the ability to specify the threshold and rationale in the data file, which can also be described in the methods in their scientific publications. We recommend that these additional user-defined flags also consider other data quality concerns. For example,

suddenly varying pitch, roll, or heading measurements may indicate unwanted movement of the instrument, or pressure sensors may indicate whether the instrument is out of the water.

### 4.1.1   Despiking velocities

Short-lived transient spikes may contaminate velocity measurements, often only a few samples in duration but with different amplitude to the neighbouring "correct" signal. It is critical to remove as many of these spikes as possible, as they can dra-

matically alter the velocity spectra, which are required for fitting the inertial subrange model. Spikes in ADV measurements may manifest because of aliasing of the Doppler signal, which happens when pulses are contaminated through reflection off complex objects and boundaries (Goring and Nikora, 2002).

To despike ADV velocity measurements, we recommend the median filter-based method described by Brock (1986). This method was originally developed for atmospheric measurements and was also recommended by Starkenburg et al. (2016)

review for despiking high-frequency atmospheric measurements of carbon dioxide used to quantify vertical turbulent fluxes. They found that the median filter was more robust than other filters such as the commonly applied phase-space thresholding method developed by Goring and Nikora (2002). In particular, the median filter method (i) is better at handling missing points, (ii) copes with the presence of low-frequency coherent turbulent structures, and (iii) is less biased by spikes than other methods (Starkenburg et al., 2016).

The method requires a window length, over which to calculate the median, and a threshold for spike identification. The window length must be longer than the duration of spikes, which can span consecutive samples but also must be sufficiently short to compute a reasonable local median (see Table 1 of Starkenburg et al., 2016). For the spike-identification threshold, we recommend a method that calculates a histogram of velocity differences between the original minus the smoothed velocities (Brock, 1986). In this technique, the local minimums on either side from the center define the positive and negative thresholds,

and velocity differences exceeding these thresholds are deemed to be spikes (Figure 4).



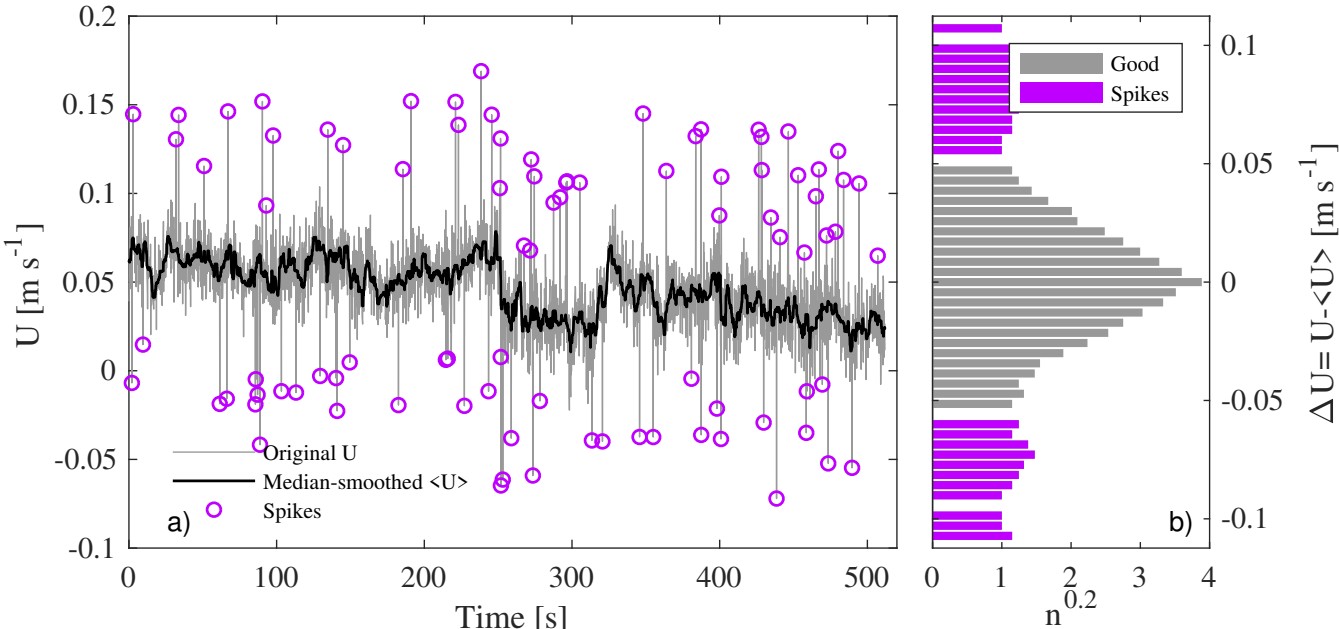

**Figure 4.** Example of the despiking procedure adapted from Brock (1986). (a) original velocities and smoothed signal obtained by applying a median filter over $2f_s+1$ samples. (b) Histogram of the difference between the original and smoothed velocity timeseries during the first despiking iteration. A power of 0.2 was applied to the number of samples $n$ in each histogram bin. The $\Delta U = 0$ threshold for identifying spikes was determined from this histogram as the first instance where $n = 0$ above and below $\Delta U = 0$. In this example, the thresholds for accepting velocities as good is $-0.056 \text{ m s}^{-1} < \Delta U < 0.051 \text{ m s}^{-1}$. If no bins have $n = 0$, then the $\Delta U$ corresponding to the minimum $n$ is used for setting despiking thresholds.

### 4.1.2 Formation of data quality flags

Each step flags velocity samples that do not meet the quality-control criteria. Each criterion uses binary (1/0) values that are then transformed into a "bitwise" flag. The maximum flag value depends on the number of criteria $N_c$ used to assess data quality. The overall value of the boolean flag increases as the number of failed quality-control metrics increases. The maximum

possible value for the flag will be less than $2^{N_c}$, which translates to 255 when eight quality-control metrics are being evaluated ($N_c = 8$).

To flag the raw velocities, we use an 8-bit boolean number calculated from:

$$\text{VEL\_FLAG} = \sum_{j \in I}^{N_c} 2^{j-1}, \tag{8}$$

where $j$ is the flag number amongst those applicable to each velocity sample. For example, $I = \{2, 5\}$ if the second and fifth

quality-control flags apply to the velocity samples, translating to a boolean value of 18. These boolean flags allow tracking the state of multiple metrics simultaneously while reporting only one number. A value of 0 means the velocity sample was





not flagged and thus is likely high-quality. These boolean flags were designed with the Climate and Forecast (CF) metadata conventions in mind.

Flags are identifiable in the NetCDF file by the FLAG suffix appended to the variable name, i.e., XYZ_VEL_FLAG for raw velocities collected in the instrument's XYZ coordinates, or EPSI_FLAG for $\hat{\varepsilon}$ estimates (Table A1). We also included additional metadata in the variables' attributes. We added "flag thresholds" and "flag thresholds meanings" to CF recommended "flag_meanings" and "flag_masks" attributes (see Table 2). The "flag thresholds" provide the thresholds associated with a given flag, while the "flag thresholds meanings" briefly explain how the thresholds are applied. These extra attributes allow future users to revisit the flags, particularly the thresholds used, and apply more stringent (or less stringent) thresholds at their

discretion.

**Table 2.** Summary of the different boolean flags and thresholds used to flag raw velocity samples. This information is stored with the suffix _FLAGS appended to the velocity variable name at the first processing level (Table A1). None of the MAVS datasets were flagged at level 1. As contamination from vortex shedding impacts the quality of $\hat{\varepsilon}$ estimates, these data are flagged at level 4.

| Flag meanings | Flag masks | Flag threshold | Flag threshold meaning |
|---|---|---|---|
| 1. Low signal to noise ratio | 1 | 10 db | Ratio is below the threshold. |
| 2. Low amplitude | 2 | 60 counts | Amplitude is below the threshold. |
| 3. Low correlation | 4 | 70 | Correlation is below the threshold. |
| 4. Obstructed beam | 8 | 40 | Amplitude difference between adjacent beams is above the threshold. |
| 5. Spike | 16 | Sampling rate | Median filter half-width number of samples (Brock, 1986). Set to the sampling rate of the instrument. |
| 6. Suspiciously large velocity | 32 | 3 | Sample exceeds the standard deviation by the set multiplying factor. |
| 7. User-defined | 64 | n/a | User can optionally flag out-of-water samples, broken probes, etc. |
| 8. Phase-wrapped | 128 | n/a | Replaced by unwrapped velocity |
| Total | 255 if all eight metrics are used | | |

## 4.2 Segmented timeseries

At Level 1, quality control of the raw velocity measurements was completed. For the spectral calculations required to compute $\varepsilon$, (1) the measurements must be segmented in time, (2) flagged data must be replaced, (3) the velocities must be rotated into the frame of reference of the mean flow, and (4) the rotated velocities must be detrended to calculate the velocity fluctuations

in the mean, transverse, and vertical directions.




The time series must be segmented before computing properties of turbulence. Selecting the length of the segments requires a balance between using sufficient data to resolve the inertial subrange in wavenumber space, while still ensuring stationarity of the turbulence. Stationarity implies that the key properties of the flow do not change over timescales shorter than the length of the segment. For many aquatic systems, this time scale is on the order of 5-15 minutes. If burst sampling is used because 260 continuous sampling is unfeasible for the duration of the deployment, the segmenting considerations discussed below should be incorporated into the experimental design, as each burst is typically considered a segment. However, it is possible to break up bursts into segments if they are long enough (e.g., Tidal slough ADV benchmark in Table 1). These choices are made at the data processing stage for continuous time series.

The choice of segment length $\tau_\varepsilon$ for estimating the dissipation impacts the resolvable wavenumbers of the inertial subrange 265 and ultimately the statistical accuracy of the final spectrum to be computed. As described below (§ 4.3), a fast Fourier transform (FFT) is used to convert the time series into frequency space. The number of velocity samples $N_{\mathrm{FFT}}$ included in each FFT, the sampling frequency $f_s$, and the mean velocity $\bar{U}$ dictate the spectral resolution and the smallest resolvable wavenumbers $\Delta\tilde{k}$:

$$\Delta\tilde{k} = \frac{f_s}{N_{\mathrm{FFT}}\bar{U}} = \frac{1}{\tau_{\mathrm{FFT}}\bar{U}}. \tag{9}$$

The duration of each FFT-length is given by $\tau_{\mathrm{FFT}} = N_{\mathrm{FFT}}/f_s$. The low wavenumber and resolution that resolves theoretically 270 at least one decade of the inertial subrange is given by:

$$\Delta\tilde{k} \lesssim \frac{1}{70L_k}. \tag{10}$$

When a decade is resolved, ten spectral samples $N_s$ are available for spectral fitting (Figure 5b). This wavenumber resolution depends on the $\varepsilon$ (Figure 5a), while the FFT duration $\tau_{\mathrm{FFT}}$ depends also on the mean speed past the sensor (Figure 5c). Increasing the $\tau_{\mathrm{FFT}}$ by a factor of 10 allows for 100 spectral samples to be theoretically resolved in the inertial subrange (Figure 275 5b). Once a $\tau_{\mathrm{FFT}}$ is chosen, the total duration of each segment (or burst duration) for estimating dissipation $\tau_\varepsilon \geq 2\tau_{\mathrm{FFT}}$, and preferably $\tau_\varepsilon \geq 3\tau_{\mathrm{FFT}}$. These choices ensure the spectra have sufficient statistical certainty (degrees of freedom) for fitting (Bluteau, 2025).

With real observations, we recommend first selecting $\tau_{\mathrm{FFT}}$ from Figure 5c using the lowest expected $\varepsilon$ and a relatively low $\bar{U}$ derived from the observations. The spectra can then be estimated and plotted in wavenumber space against the theoretical 280 velocity spectra in a similar format as Figure 2. This visual representation can immediately show whether the chosen $\tau_{\mathrm{FFT}}$ is longer than necessary (e.g., Tidal Shelf High Quality example in Figure 2) or if the length needs to be extended because the high wavenumbers are drowned by noise. The goal is to choose a $\tau_{\mathrm{FFT}}$ that is sufficiently long to resolve a decade of the inertial subrange throughout the entire dataset. The duration $\tau_{\mathrm{FFT}}$ will be longer than necessary when $\varepsilon$ or $\bar{U}$ increases (Figure 5c). For our benchmarks, the total segment duration varied from 82 seconds for the high-energy Tidal MAVS benchmark to 285 1024 seconds for the low-energy Under-ice MAVS benchmark (Table 1).

The Level 1 flags can exclude data points from further analysis. Once the time series has been segmented, data loss due to flagged points must be addressed before spectral calculations. For spectral computations, these excluded data points must be replaced in the time series as appropriate. It is recommended that linear interpolation replace missing points and record the





**Figure 5.** a) Minimum FFT length scale $\ell_{\mathrm{FFT}}$ that must be resolved by the spectra as a function of $\varepsilon$ and the number of fitted spectral samples $N_s$. The lowest resolved wavenumber and resolution $\Delta\tilde{k} = \ell_{\mathrm{FFT}}^{-1}$. (b) The number of spectral samples available for fitting as a function of the number of resolved decades $\delta = \log_{10}\left(\frac{k_{is}}{\Delta\tilde{k}}\right)$ within the inertial subrange. (c) Minimum FFT-length duration $\tau_{\mathrm{FFT}}$ required for resolving $N_s = 10$ spectral samples within the inertial subrange. Note (a) and (c) represent minimum lengths and durations since the available bandwidth for spectral fitting may be reduced because of measurement noise and/or anisotropy.

percent of good samples in each segment in the NetCDF Level 2 and 4 data. Segments with more than 10% missing data should
290 be flagged and rejected (Table 3). The threshold chosen for rejection should be recorded at Level 4 in the NetCDF file within
the EPSI_FLAGS metadata (Table A4).

To estimate $\varepsilon$ from all the different velocity components, the measurements must be rotated into the main direction of the
flow. In some instances, the instrument's frame of reference may be aligned with the direction of flow, which is ideal to account





for the varying levels of anisotropy among components (Gargett et al., 1984; Bluteau et al., 2011). If this alignment isn't set,
then the velocities' measurement frame must be rotated into that of the flow, which we refer to as the analysis frame of reference.
This process can be done by using the time-averaged velocities in each segment. If only the vertical velocity component will
be used for calculation $\varepsilon$, then this step may not be necessary. The frame of reference for velocity analysis is noted in the
NetCDF metadata within the Level 2 hierarchal group (Table A2). These velocities should be stored at Level 2 in a 3-D matrix
UVW_VEL with dimensions [N_SEGMENT, N_SAMPLE, N_VEL_COMPONENT] with the rotation method (if any) for
obtaining UVW_VEL velocities noted in the Level 2 metadata. Each N_SEGMENT are associated with a unique timestamp
taken at the mid-point of each segment and stored in the TIME variable. We also store at Level 2 the variables ROT_AXIS
and ROT_ANGLE, which can be used for rotating velocities from the measured coordinate system (e.g., XYZ_VEL) into the
analysis frame of reference UVW_VEL (Table A2). This information is helpful for recovering the velocities of each segment
in the original frame of reference.

## 4.3 Spectral observations

The parameters chosen when computing spectra can restrict the range of resolved wavenumbers, thus impacting the suitability
of the spectra for estimating dissipation rates $\hat{\varepsilon}$. In particular, spectra must resolve as much of the available inertial subrange
as possible (Figure 2) while considering how measurement noise may dominate the high wavenumbers close to the viscous
subrange, which tend to be more isotropic than low wavenumbers. These choices also impact the statistical reliability of the
velocity spectrum, and thus $\hat{\varepsilon}$ obtained from fitting algorithms as well as the accuracy of the spectral slope estimates (Bluteau,
2025) — a quality-control indicator presented below in §4.6.

Our recommended spectral averaging involves splitting the time series into $N_f$ subsets that are 50% overlapped and win-
dowed using a Hanning function (Lueck et al., 2024). An FFT is applied to each windowed subset before computing the squared
magnitude to get the power spectral density estimates (chapter 8 for pwelch methods; Percival and Walden, 2020). These power
spectral density of all subsets are then averaged together to yield the spectra used for estimating $\varepsilon$. When computing spectra
from 50% windowed time series with a Hanning (cosine) window results in degrees of freedom $d$:

$$d = cN_f = c\left(\frac{2\tau_\varepsilon}{\tau_{\text{FFT}}} - 1\right) \tag{11}$$

with $c = 1.9$ (Equation 416 of Percival and Walden, 2020) or $c = 1.92$ according to Nuttall and Carter (1980). With our
suggested segment duration of $\tau_\varepsilon \geq 3\tau_{\text{FFT}}$ (§4.2), $N_f \geq 5$ since $N_f = 2\tau_\varepsilon/\tau_{\text{FFT}} - 1$. The resulting spectral observations will
have about 10 degrees of freedom, i.e., $d \approx 10$. This recommendation is based on log-fitting methods returning $\hat{\varepsilon}$ within a factor
of about 1.5 of the actual value when applied to synthetic spectra with $d = 10$ (Bluteau, 2025)

## 4.4 Estimates of turbulent kinetic energy dissipation $\varepsilon$

We now discuss obtaining $\varepsilon$ from the spectral observations, which involves spectral fitting Equation 2 to the wavenumbers
within the inertial subrange. We will focus first on the fitting methods before addressing how to identify the wavenumbers that
belong to the inertial subrange, as these wavenumbers depend on $\varepsilon$ – the sought quantity.





### 4.4.1  Spectral fitting techniques

We considered six methods to fit the model $\Psi_j(k)$ to the spectral observations $\Phi_j(k)$, and subsequently estimate $\varepsilon$. The details of the methods and their assessment against synthetic spectra are described in Bluteau (2025). Spectra were synthesized by specifying $\varepsilon$ in Equation 2 and adding uncertainty to the spectra via two different statistical distributions. The sensitivity of the fitting methods was also evaluated against the uncertainty (smoothness) of the spectral observations and the number of samples used in the fitting.

From this analysis, we recommend using fitting methods applied to log-transformed spectral observations. Specifically, we recommend minimizing the least-absolute deviation (residuals) between the fitted model and the observations. This method requires no assumption about the statistical distribution of the observations, unlike least-square regression, which expects normally distributed data. Minimizing the absolute residuals is considered less stable than least-square regression (Tercan, 2021). However, since only the intercept $\beta_0$ that best fits the log-transformed spectral observations $\hat{\Psi}(k)$ is required; the technique amounts to:

$$\beta_0 = \text{median}\left[\ln(\hat{\Psi}_i) - \beta_1 \ln(k_i)\right]. \tag{12}$$

The spectral slope $\beta_1$ is set to the expected value of $-5/3$ for the inertial subrange (Equation 2), and $i$ denotes each observation in the spectra. Linear least squares would take the mean of Equation 12 rather than the median. Both least-square regression and least-absolute deviation performed well against the synthetic spectra (Bluteau, 2025). However, the estimated $\hat{\varepsilon}$ from least-absolute deviation were less biased, especially for spectra with low degrees of freedom, i.e., high degrees of uncertainty because of limited spectral averaging.

From the fitted intercept $\hat{\beta}_0$, we can estimate $\hat{\varepsilon}$ using:

$$\hat{\varepsilon} = \left(\frac{\exp(\beta_0)}{a_j C_k}\right)^{3/2}, \tag{13}$$

where $a_j C_k$ are the constants already defined above for the inertial subrange model (Equation 2). Using the ladLog fitting method and FFT-lengths that are 1/4 of the segment length ($d \approx 14$), the estimated $\hat{\varepsilon}$ is expected to be within 43% of the actual $\varepsilon$ if ten samples are fitted over one decade (Figure 3 and 4 of Bluteau, 2025). This error reduces to 15% when the number of samples increases to 100, which tends to occur when the fit falls at wavenumbers nearing the high wavenumber limit of the inertial subrange. The synthetic spectra were also used to evaluate the ability of the fitting technique to estimate the spectral slope $\hat{\beta}_1$ (Bluteau, 2025). The spectral slope estimates help flag spectra that do not exhibit a clear inertial subrange because of poor data quality, low energy, and anisotropy, or simply because the sampling protocol or spectral averaging cannot resolve the inertial subrange. The latter occurs when sampling too slowly or using fft-lengths that are too short to resolve the entire inertial subrange (§4.3). As with the estimation of $\hat{\varepsilon}$, the methods applied to the log-transformed data were better at recovering the spectral slope $\hat{\beta}_1$ than those used to untransformed data. The logLAD method was less sensitive to outliers than the least-square regression (Bluteau, 2025).

Their results from fitting synthetic spectra using the logLAD method are shown in Figure 6 to determine a relationship for flagging $\hat{\varepsilon}$ estimates. The deviation of $\hat{\beta}_1$ from the expected $\beta_1 = -5/3$ varied with the number of spectral samples fitted $N_s$,



the degrees of freedom $d$ of the synthetic spectra, and the decadal range:

$$\delta = \log_{10}\left(k_N/k_1\right) \tag{14}$$

between the first and $N_s$ fitted wavenumber. The estimated 99.7% bounds from fitting synthetic spectra can be mathematically represented by:

$$\hat{\beta}_1 > \beta_1 \pm \frac{A}{\delta\sqrt{2dN_s}} \tag{15}$$

with $A$ ranging between 7 to 17 (Figure 6). A small $A$ is a stricter threshold for deeming spectra as exhibiting an inertial
subrange. This factor is documented in the metadata of the Level 4 NetCDF quality-control $\hat{\varepsilon}$ flags (EPSI_FLAGS in Table A4). To apply Equation 15, we also store at Level 4 the number of samples fitted $N_s$ as N_FITTED, the bounds of fitted wavenumbers K_BNDS to calculate $\delta$, and the degrees of freedom $d$ (Table A4).

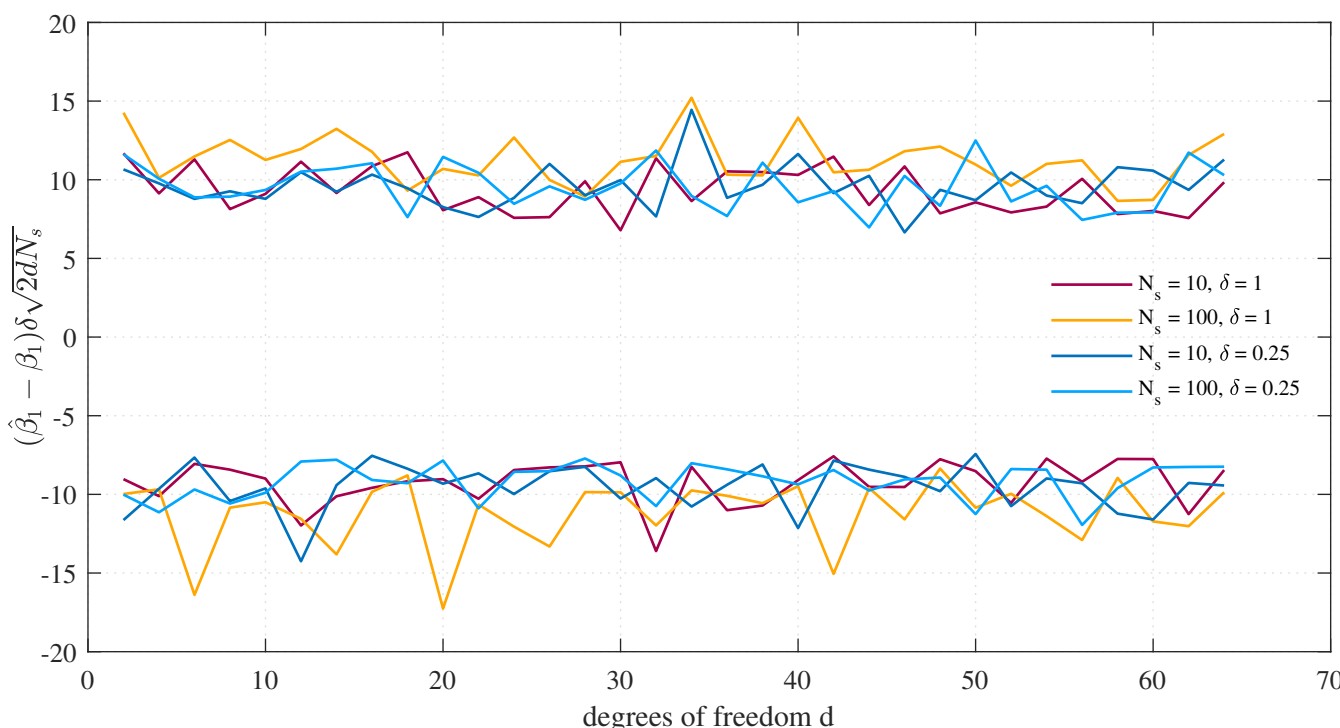

**Figure 6.** (a) The 99.7% bounds (0.15$^{th}$ and 99.85$^{th}$ percentiles) for the logLAD method and the $\chi_d^2$ distributed synthetic spectra dataset. The numerical experiments are shown for both numbers of samples $N_s$ and decadal range $\delta$ fitted by Bluteau (2025). Results are scaled by the expected standard deviation of $\chi_d^2$ distributed samples ($\sqrt{2d}$) in addition to $\delta\sqrt{N_s}$. This figure illustrates that deviation of $\hat{\beta}_1$ from the true value $\beta_1 = -5/3$ scales approximately with $\delta\sqrt{2dN_s}$.





### 4.4.2 Locating the inertial subrange in spectral observations

A primary challenge in obtaining $\hat{\varepsilon}$ is identifying the spectral wavenumbers that most likely belong to the inertial subrange,

as these wavenumbers depend on $\hat{\varepsilon}$ – the sought quantity. With appropriate choices for data sampling, segmenting (§4.2), and spectral averaging (§4.3), this step amounts to avoiding the low wavenumbers dominated by anisotropy and the high wavenumbers dominated by instrument noise. Other wavenumbers to avoid are those impacted by vortex shedding off instrument frames (e.g., Under-ice MAVS in Figure 2), or surface waves.

We evaluated four strategies for identifying the inertial subrange within the spectra by adding white noise to the synthetic of

Bluteau (2025). Two strategies involved taking the absolute deviation between the log-transformed spectra and the model and taking the mean or median of the quantity $|\ln \hat{\Psi} - \ln \Psi|$. The third strategy estimated the the mean absolute deviation of $\hat{\Psi}/\Psi$ (Equation 24 of Ruddick et al., 2000). Our fourth and recommended strategy is to estimate $\hat{\varepsilon}$ and the spectral slopes $\hat{\beta}_1$ over short wavenumber subsets of the spectra (see Figure 7). The wavenumbers with the estimated slope closest to the expected $\beta_1 = -5/3$ for the inertial subrange are then selected to calculate the spectrum's final $\hat{\varepsilon}$.

In practice, the user must select the size of each subset, in addition to the wavenumber overlap for each subset within the spectra. For our benchmark datasets, we used an overlap equivalent to 1/20 of a decade, but other users may prefer shifting the window by one spectral sample at a time. We recommend a minimum decadal wavenumber range of $\delta = 0.8$ and that each fitted subset includes at least ten samples, especially when $\delta < 1$ (Bluteau, 2025). Users may also specify a maximum $k_{\max}$ and minimum wavenumbers $k_{\min}$ that can be fitted upon inspection of the spectral observations (e.g., Figure 2). For example,

$k_{\max} \approx 2\pi/L_b$ where $L_b$ is the distance to the nearest boundary, while $k_{\min} \approx \pi/\ell$ depends on the dimension $\ell$ of the sampling volume of the instrument. For each subset, the estimated $\hat{\varepsilon}_i$ is used to calculate $L_K$, and verify whether the fitted wavenumbers are within the inertial subrange. The user may provide some allowance, but we recommend ensuring that the median fitted $k_{\mathrm{med}}$ is within the inertial subrange, i.e., $k_{\mathrm{med}} < 0.1/L_K$. Spectra for which none of the fitted subsets satisfy this requirement were flagged (see § 4.6).

### 4.4.3 Impact of noise on $\varepsilon$ estimates

In some instances measurement noise may adversely impact the estimated $\hat{\varepsilon}$, and render the spectra unusable for estimating turbulence quantities. The drowning of the inertial subrange by noise is particularly common in low-energy environments such as the ocean interior or in lakes. To deal with this issue, some authors have presumed the shape of the noise spectra to either remove it from the spectral observations (e.g., Davis and Monismith, 2011), or add the presumed noise shape to the model

used for fitting the unaltered spectral observations (e.g., Ruddick et al., 2000). Rather than use these strategies, we recommend comparing the estimated $\hat{\varepsilon}$ to the minimum $\varepsilon_m$ when the corresponding theoretical spectral energy levels in the inertial subrange (Equation 2) exceeds the noise floor $\Phi_n$ (Bluteau et al., 2011):

$$\varepsilon_m = \left(\frac{\Phi_n}{a_j C_k}\right)^{3/2} k^{5/2}. \tag{16}$$





**Figure 7.** Example of locating the inertial subrange from (a) the spectra of two velocity components collected concurrently in the *Tidal Shelf ADV*. (b) Deviation of the estimated $\hat{\varepsilon}$ from subsets, including a decade's worth of spectral samples from the $\varepsilon_{\text{best}}$ estimate, which represents the $\hat{\varepsilon}$ that yields the spectral slope $\hat{\beta}_1$ closest to the expected -5/3 presented in panel (c). The top secondary x-axis in (a) shows the non-dimensional wavenumber based on the best-fit $\varepsilon$ from the vertical, less noisy component.



This recommendation is based on the relatively low positive bias introduced when leaving the noise "as-is" in the spectra
(Figure 8). The noise floor $\Phi_n$ typically appears as white noise (flat) in velocity spectra and can be estimated by averaging
the energy levels of the highest wavenumbers where $\hat{\beta}_1 \approx 0$. Equation 16 depends on the fitted wavenumbers $k$ and can be
generalized further as a function of the smallest turbulent length scales $L_k$ defined in Equation 3 (Bluteau et al., 2011):

$$k = \frac{\alpha}{L_k} = \frac{\alpha}{(\nu^3/\varepsilon)^{1/4}}. \tag{17}$$

The inertial subrange corresponds to $\alpha \leq 0.1$. Substituting this relationship for $k$ into Equation 16 yields the following mini-
mum $\varepsilon_m$:

$$\varepsilon_m = \frac{\alpha^{20/3}}{\nu^5} \left( \frac{\Phi_n}{a_j C_k} \right)^4. \tag{18}$$

This equation demonstrates that noise is most detrimental when fitting wavenumbers approaching the theoretical high wavenum-
ber bound of the inertial subrange ($\alpha = kL_k = 0.1$). The minimum resolvable $\varepsilon_m$ can be viewed as the measurement detection
limit. The limit increases with noise levels, albeit the increase is amplified when when fitting large wavenumbers i.e., $k$ close
to the beginning of the viscous subrange (Figure 8). For example, the inertial subrange sits above the noise floor for $\varepsilon > 10^{-7}$
W kg$^{-1}$ at wavenumbers ten times smaller than the highest within the inertial subrange ($\alpha = 0.01$) provided the noise levels
$\Phi_n$ are less than $10^{-6}$ m$^2$ s$^{-2}$/(rad/m) (Figure 8a). However, if fitting the highest wavenumbers within the inertial subrange
($\alpha = 0.1$), the spectra sit above the noise floor only if $\Phi_n \lesssim 2 \times 10^{-8}$ m$^2$ s$^{-2}$/(rad/m) (Figure 8b).

By leaving the noise floor "as-is" in the spectral observations, a positive bias occurs when estimating $\hat{\varepsilon}$. The over-estimation
is in addition to other errors associated with spectral fitting techniques described above. When $\hat{\varepsilon} \approx \varepsilon_m$, the estimated $\hat{\varepsilon}$ over-
predicts the prescribed value $\varepsilon$ by a factor of 3 (circles in Figure 8). When $\hat{\varepsilon} \approx 100\varepsilon_m$, the estimated $\hat{\varepsilon}$ over-predicts $\varepsilon$ by a factor
of 1.5 (squares in Figure 8). These positive biases are relatively small compared to the range of $\varepsilon$ encountered in environmental
flows, especially considering how quickly the bias lessens when including lower wavenumbers ($\alpha \leq 0.1$) in the fit. Hence, the
biases shown in Figure 8 are conservative when $\alpha$ is determined from the largest fitted wavenumber $k_N$.

To flag overly noisy spectra with Equation 18, the user must estimate the spectral noise floor $\Phi_n$ from the observations.
For our benchmarks, $\Phi_n$ was calculated as the average spectral energy levels over the last $\delta = 0.1$ decadal range or the last 30
spectral samples, whichever leads to a larger number of averaged samples. The resulting $\Phi_n$ is stored as SPEC_NOISE_UVW
in the same units as the spectra from which the average was determined at Level 3 UVW_VEL_SPEC. The spectral noise
estimate $\Phi_n$ is used with the maximum fitted wavenumber $k_N$ stored in K_BNDS and the estimated dissipation $\hat{\varepsilon}$ stored in
Level 4 to estimate $\varepsilon_m$ using Equation 18. The estimated $\varepsilon_m$ is stored as MIN_EPSI_NOISE at Level 4 so that this value can
be used to generate quality control flags for $\hat{\varepsilon}$ in section 4.6.

### 4.4.4 Flow interference

Nearby instrument frames may obstruct the flow or shed vortexes, contaminating the velocity observations. This contamination
is often recognizable as a narrow band peak in the velocity spectra when the frame obstruct the flow upstream of the sampling







**Figure 8.** Over-estimation of $\hat{\varepsilon}$ as a function of noise level $\Phi_n$ and wavenumber fitted. The fitted wavenumber in (a) corresponds to $\alpha = 0.01$; in (b), to $\alpha = 0.1$, the high $k$ limit for the inertial subrange. The circles show the resolvable dissipation $\varepsilon_m$ (Equation 18). The dashed line indicates that the inertial subrange sits below the noise floor; thus, $\hat{\varepsilon}$ cannot be resolved.

430 following (Figure 2). In this situation, the contamination frequency $f_c$ can be determined from the Strouhal number $Sr$:

$$f_c = 0.21 \frac{\bar{U}}{D} \tag{19}$$

for high Reynolds flow ($Sr \approx 0.21$, Kundu, 1990). The frequency of the disturbances increases with decreasing diameter $D$ of the obstruction or an increase in the velocities past the frame $\bar{U}$. These disturbances are typically associated with a specific flow direction relative to the instrument's frame of reference. Hence, if the instrument is fixed in space, the flow direction can

435 be used to identify when vortex shedding is potentially contaminating the velocity spectra, Alternatively, near boundaries, the





estimated $\hat{\varepsilon}$ can be compared to the predicted values for the log-law of the wall and the flow direction (see Figure 4 McGinnis et al., 2014). This contamination may occur at wavenumbers higher than those within the inertial subrange, as apparent in the first 25 minutes of the *Tidal Shelf ADV* benchmark. The flow disturbances will sometimes be significant enough to increase energy levels within the inertial subrange. Thus, it is preferable to avoid the problem by placing the sampling volume of the instruments far from the obstruction. A common rule of thumb is leaving a space larger than ten times the diameter of the obstruction between the frame and the sampling volume. When the sampling volume is obstructed, $\hat{\varepsilon}$ will be over-estimated downstream of the disturbance (see Figure 4 of McGinnis et al., 2014).

Another contamination source specific to the MAVS velocity sensor is the rings that house the acoustic transducers. These rings shed vortexes, contaminating the velocity measurements. This contamination affects most directions transverse to the instrument's main shaft. For a horizontally mounted instrument, such as our *Tidal MAVS* benchmark, the longitudinal direction was the least affected by the rings' vortex shedding (Hay et al., 2013). In contrast, the transverse direction was the most affected. The other *Under-ice MAVS* benchmark was mounted vertically on a rod lowered beneath the ice sheet. The vertical velocity direction was the least affected, followed by the longitudinal, although the vertical component still shows evidence of a high-frequency flow disturbance in its spectra (Figure 2). Our quality-control metrics below include optional flagging for $\hat{\varepsilon}$ estimates affected by flow disturbances.

### 4.5 Confidence intervals for $\hat{\varepsilon}$

Given the above recommendation for fitting the spectra using the least-absolute deviation method, we suggest creating confidence intervals on $\hat{\varepsilon}$ using bootstrapping techniques (Davison and Hinkley, 1997). The advantage of bootstrapping is that no assumption is made about the statistical distribution of the observations. Bootstrapping involves resampling the dataset and recomputing the desired statistics to provide a distribution of estimates, from which confidence intervals can be obtained. For our application, we bootstrapped the fitted $\hat{\beta}_0$ since it is used to estimate $\hat{\varepsilon}$ from Equation 13. This step is achieved by resampling the residuals $r$ between the log-transformed observations ($\ln \hat{\Psi}$) and the best-fit $\hat{\beta}_0$:

$$r_i = \ln \hat{\Psi}_i - \left( \hat{\beta}_0 - \frac{5}{3} \ln k_i \right) \tag{20}$$

and adding them to the best-fit line in log-log space to create a new dataset $\ln \tilde{\Psi}$:

$$\ln \tilde{\Psi}_i = \tilde{r}_i + \left( \hat{\beta}_0 - \frac{5}{3} \ln k_i \right), \tag{21}$$

where $i$ represents an observation and $\tilde{r}_i$, the bootstrapped residuals. The new spectral log-transformed spectra $\ln \tilde{\Psi}$ are refitted with Equation 12 to obtain a bootstrapped $\hat{\beta}_0$ estimate. The resampling of residuals and fitting are repeated many, typically more than 1000, times (Davison and Hinkley, 1997), to obtain a collection of bootstrapped $\hat{\beta}_0$ estimates for a given velocity spectrum, and thus $\hat{\varepsilon}$ estimate. Finding the 2.5[th] and 97.5[th] percentiles from the collection of $\hat{\beta}_0$ provides the 95% confidence interval. These percentiles are then substituted into Equation 13 to obtain the confidence levels for $\hat{\varepsilon}$. In the benchmark datasets, we obtained the 95% confidence intervals for each $\hat{\varepsilon}$ estimate by resampling the residuals and refitting 1000 times $\ln \tilde{\Psi}$ (Equation 21).



## 4.6 Quality-control considerations and flags

Similarly to the raw velocities at Level 1, we assess our $\hat{\varepsilon}$ estimates against multiple quality-control metrics. The results of this

assessment are stored using an 8-bit ($N_c = 8$) boolean flag calculated with the same equation as for our raw velocities (eq. 8). Thus, the maximum flag value is 255 when all eight quality-control metrics apply to a given $\hat{\varepsilon}$ estimate. This number is stored at Level 4 as EPSI_FLAGS. Below we describe each metric individually, which are summarized in Table 3.

**Table 3.** Summary of the different boolean flags and thresholds used to mask $\varepsilon$ estimates, stored as EPSI_FLAGS variable in the 4th processing level within NetCDF file.

| Flag meanings | Flag masks | Flag threshold | Flag threshold meaning |
|---|---|---|---|
| 1. Non-stationary | 1 | 20 subsets | Number of subsets used for calculating runs. The acceptable number of runs is in between 6 and 15 when subdividing the dataset into 20 subsets. |
| 2. Failed Taylor hypothesis | 2 | $\mathcal{T}_H$ in Eq. 22 | Ratio is above the acceptable threshold $T_H = 0.33$ (Pécseli and Trulsen, 2022). |
| 3. Noise dominated spectra | 4 | $\mathcal{T}_n$ in Eq. 23 | Ratio $\varepsilon/\varepsilon_m$ is below the acceptable threshold. A low ratio implies the high wavenumbers of the inertial subrange are drowned by noise. |
| 4. Poor spectral slope | 8 | $A$ in Eq. 15 | Smaller $A$ reduces the range of acceptable slopes $\hat{\beta}_1$ for an inertial subrange. |
| 5. Missing velocity samples | 16 | $\mathcal{T}_P = 10\%$ | Maximum permissible percentage of missing velocity samples in a segment. |
| 6. Anisotropic (optional) | 32 | $\mathcal{T}_A$ in eq. 24 | Ratio of largest to smallest turbulent overturns too low for the spectra to exhibit a well-defined isotropic inertial subrange. May require information about the background shear and/or stratification, especially if far from a boundary. |
| 7. Outside inertial subrange | 64 | Eq. 25 | Estimated $\hat{\varepsilon}$ places the fitted $k$ within the viscous subrange. |
| 8. User-defined | 128 | | Example of user-defined metrics could be outliers in the $\hat{\varepsilon}$ timeseries, contamination from vortex shedding shedding, etc. |
| Total | 255 if all eight metrics are used | | |



### 4.6.1 Non-stationarity

Our first metric presents the results from testing the stationarity of turbulent velocities. Both spectra computations and the
inertial subrange model rely on the assumption of stationarity. The test is applied to each velocity component separately. For
this purpose, we used the nonparametric test by Bendat and Piersol (2000), which involves calculating two statistics (standard
deviations and average) over shorter subsets of each velocity timeseries segment (of duration $\tau_\varepsilon$). We then compare the number
of runs (crossings) of each statistic about its average to the expected values in Table A.6 of Bendat and Piersol (2000). For
this evaluation, the user must choose the significance level and the number of subsets to subdivide each segment. We used 20
subsets and a 95% level, resulting in an acceptable number of runs between 6 and 15. We flagged $\hat{\varepsilon}$ as non-stationary if the
number of runs for the standard deviation and means for the velocity segment are outside the expected range.

### 4.6.2 Violation of Taylor's frozen hypothesis

The second quality metric focuses on Taylor's frozen turbulence hypothesis. We recommend flagging $\hat{\varepsilon}$ estimates associated
with low advection velocities $\bar{U}$ relative to the root mean square of the turbulent velocity fluctuations along the direction of
mean advection $u_{\mathrm{rms}}$. This condition translates to flagging $\hat{\varepsilon}$ associated with $u_{\mathrm{rms}}/\bar{U}$ exceeding a threshold $\mathcal{T}_H$:

$$\frac{u_{\mathrm{rms}}}{\bar{U}} > \mathcal{T}_H. \tag{22}$$

We suggest $\mathcal{T}_H = 0.33$ based on the work of Pécseli and Trulsen (2022) who showed that the error for $\hat{\varepsilon}$ for this value is
about 10%. Larger errors are expected when $\frac{u_{\mathrm{rms}}}{\bar{U}}$ increases (see Figure 12 of Pécseli and Trulsen, 2022). Thus, $\mathcal{T}_H$ could be
increased but it should always remain smaller than 1. The chosen threshold should be specified in the metadata of EPSI_FLAGS
accompanying the $\hat{\varepsilon}$ estimates as detailed in Table 3.

### 4.6.3 Noise-dominated spectra

The third flag considers whether the fitted spectra are drowned by noise. This criteria involves calculating the minimum resolv-
able $\varepsilon_m$ using equations 17 and 18 from the dissipation $\hat{\varepsilon}$ estimates, the maximum fitted wavenumber $k_{\mathrm{max}}$, and the estimate
of the spectral noise floor $\Phi_n$ (see § 4.4.3). The estimated $\hat{\varepsilon}$ is compared to the minimum resolvable $\varepsilon_m$. Dissipation estimates
are flagged as being drowned by noise when the ratio between these two quantities is less than the user-defined threshold $\mathcal{T}_n$:

$$\frac{\hat{\varepsilon}}{\varepsilon_m} < \mathcal{T}_n. \tag{23}$$

The threshold $\mathcal{T}_n$ must always be larger than one and specified in the $\hat{\varepsilon}$ flag's metadata (see Table 3). We suggest $\mathcal{T}_n = 3$ owing
to the positive bias shown when the highest wavenumbers of the inertial subrange sit near spectral observations as illustrated
in Figure 8).

### 500  4.6.4 Spectral slope outside expected range

Our fourth flag involves estimating the spectral slope $\hat{\beta}_1$ from the observations. We identify $\hat{\varepsilon}$ associated with spectral slopes
that deviate too much from the expected -5/3 value. This situation may occur because of excessive noise, anisotropy, or other





contamination (e.g., vortex shedding). Spectral slope estimates that fall outside the bounds given by Equation 15 are flagged. To flag $\hat{\varepsilon}$, we set $A = 17$ because we estimated the spectral slopes $\hat{\beta}_1$ using methods applied to log-transformed spectral

observations. This $A$ value is based on the 99.7% bounds in Figure 6. Reducing $A$ would render the threshold more stringent, flagging an increased number of $\hat{\varepsilon}$ estimates. Users should always specify their choice of $A$ in the metadata accompanying their $\hat{\varepsilon}$ estimates. This information in combination with the variables for estimating the acceptable bounds (Equation 15) should be available to enable other users to re-flag $\hat{\varepsilon}$ if desired.

### 4.6.5 Missing too many samples

Our fifth flag involves identifying segments with significant data loss during the quality-control of raw velocities, which render the spectra unreliable for estimating $\hat{\varepsilon}$ (§ 4.1). Limited testing involving the random removal of velocity samples from our benchmarks showed that spectral shapes deviate considerably from the original when more than 10% of samples are removed. As the percentage of data loss increases, the interpolated time series yield spectra with increased energy levels at low $k$ and decreased energy at high $k$. We suspect acceptable data loss depends on data quality (i.e., noise levels) and underlying turbu-

lence captured in the original time series. We thus recommend users specify their threshold $\mathcal{T}_P$ for the maximum percentage of data loss in the segment after quality-controlling the raw velocities. For our benchmark datasets, we use 10% as the minimum percentage of good samples in a segment.

### 4.6.6 Spectral anisotropy

The sixth metric is for flagging anisotropic $\hat{\varepsilon}$ estimates. This flag is optional as it typically requires an estimate of the largest

turbulent overturns $L$ (see §2). The size of the large overturns depends on the mean flow characteristics and so necessitate measuring the background stratification ($L_O$, Equation 5) or the background shear ($L_S$, Equation 6) unless the distance from the boundary is a suitable alternative for $L$ (Equation 7). Spectra are considered too anisotropic when the ratio $L/L_K$ is small, which would potentially lead to underestimating $\hat{\varepsilon}$. To flag this issue, we compare this ratio to a user-defined threshold $\mathcal{T}_A$:

$$\frac{L}{L_K} < \mathcal{T}_A. \tag{24}$$

The threshold $\mathcal{T}_A$ depends on the chosen measure for $L$ and the velocity component (see Bluteau et al., 2011, for an extensive review). The longitudinal direction tends to have a broader inertial subrange than the vertical and transverse directions. We recommend a similar threshold $\mathcal{T}_A = 100$ to Bluteau et al. (2011), noting that higher thresholds might be necessary when the transverse or vertical components are used to estimate $\hat{\varepsilon}$. The user should specify the definition of their largest $L$ and the threshold $\mathcal{T}_A$ for flagging the data in the EPSI_FLAG metadata. For our benchmarks, this length-scale was set $L = \kappa L_b$ and

stored as at level 4.

### 4.6.7 Fit located outside inertial subrange

The seventh flag identifies spectra when most fitted wavenumbers sit outside the inertial subrange. This situation arises when the median fitted wavenumber $k_{\mathrm{med}}$ during the search of the inertial subrange (see §4.4.2) are high and always lies within the





viscous subrange:

$$k_{\mathrm{med}} \gtrsim \frac{0.1}{L_K} \tag{25}$$

(Pope, 2000). This situation typically arises when the speed past the sensor is very slow, or the spectra are very noisy, such that the algorithm places the inertial subrange at very high wavenumbers.

Alternatively, the fitted wavenumbers may be too small and thus outside the inertial subrange. This situation will arise if the inertial subrange is unresolved because the sampling frequency is too slow or the largest overturns' are negligible. For the benchmarks, these wavenumbers are those smaller than those dictated by the distance to the nearest boundary (see Figure 2).

### 4.6.8 User-defined flags

The last flag is reserved for missing $\hat{\varepsilon}$ estimates or any other user-defined flag. For example, the user may flag data loss onboard the instrument, $\hat{\varepsilon}$ outliers in the time series, or perhaps unrealistically different $\hat{\varepsilon}$ between velocity components. Occasionally, all components will yield $\hat{\varepsilon}$ estimates, passing all quality control criteria, but significant differences still exist between components. This situation may occur, for instance, because of vortex shedding from nearby flow obstacles (§ 4.4.4). We used this user-defined flag to denote velocity directions from the MAVS benchmark datasets that were impacted by vortex shedding.

### 4.6.9 Final $\hat{\varepsilon}$ estimates

In the NetCDF file, a final estimate for $\hat{\varepsilon}$ is stored as a 1d timeseries EPSI_FINAL. This parameter is effectively the "best" $\hat{\varepsilon}$ issued from the data processing and flagging with the EPSI_FLAGS. There can often be large differences in dissipation estimates among the three velocity components caused by differing impacts of noise, anisotropy at low wavenumbers, and vortex shedding on the spectral observations. Thus, we select one of the velocity component to produce the final $\hat{\varepsilon}$ estimates, and document the choice in the EPSI_FINAL metadata.

## 5 Application of methods to benchmark datasets

We illustrate the methods, common data quality issues, and the application of quality-control flags for our *Tidal Shelf ADV* benchmark. The quality-control thresholds and processing choices are summarized in Table 4 for our four benchmarks. The velocities' "legged" appearance was caused by setting the velocity range below the maximum observed during deployment. This issue is rectified once beam velocities are unwrapped. These velocities were stored as XYZ_VEL_UNWRAP at level 1, and once quality-controlled, they are segmented and included in the NetCDF file at level 2 (Figure 3 and Table A1). This dataset was high quality as the data return was more than 85% for all segments (Figure 9c). The most significant data loss coincided with the period of strong flows when many velocity samples were phase-wrapped. Despite the unwrapping, some poor velocities samples remained and were flagged using the 5 and 6th flags that denote spikes and suspiciously large velocities in Table 2. These samples appeared in the timeseries as having velocities flags totaling 160 and 176, respectively (Figure 9c). For this dataset, the 8th flag, denoting phase-wrapped samples, was not used to discard velocity samples. This flag yielded a





boolean value of 128. To be discarded, the samples must be flagged for another reason, so only samples with flags greater than

565   0 and not equal to 128 were replaced using linear interpolation at level 2 (see §4.3). This interpolated dataset was then split
into 256-sec long segments with a 25% overlap between adjacent segments and stored in the Level 2 group within the NetCDF
file.

**Figure 9.** Level 1 and 2 observations associated with the *Tidal Shelf ADV* benchmark. (a) Raw velocities with obvious signs of phase-wrapping. (b) Quality-controlled and unwrapped velocities for the benchmark. (c) Maximum boolean velocity flags (i.e., XYZ_VEL_FLAG) value for each 256 s long segment. The secondary axis shows the percentage of good samples within each segment. (d) Mean velocities and direction relative to the instrument's frame of reference. Table 2 summarizes the meaning of the velocity boolean flags.





The Level 2 segmented and quality-controlled data were then used to calculate the spectra stored in the NetCDF file at Level 3. Spectra from four different segments of the *Tidal Shelf ADV* benchmark are illustrated in Figure 10. This benchmark displayed evidence of vortex shedding when velocities were directed at $220°$ relative to the instrument's frame of reference and exceeded 5 cm s$^{-1}$ (segments 1 to 7). However, the shedding was well outside the inertial subrange (100 cpm, Figure 10a), so $\hat{\varepsilon}$ was not flagged for this criteria.

The spectra from segments 8 through 10 were not impacted by vortex shedding because the flow was too weak despite the velocities being oriented in the right direction to contaminate the measurements (see segment 10 in Figure 10b). In this second example, the $\hat{\varepsilon}$ estimates of all velocity components were nonetheless flagged for failing the Taylor Hypothesis criteria and for most of the spectra sitting within the theoretical viscous subrange based on the fitted $\hat{\varepsilon}$ (2nd and 7th flags, respectively, in Table 3). Combined, these two flags translate to a boolean value of 66 for the longitudinal and transverse $\hat{\varepsilon}$ flag stored at level 4 (Equation 8). The vertical component had a larger flag of 67 because it was also deemed non-stationary (1st flag in Table 3).

**Table 4.** Data processing choices when estimating $\hat{\varepsilon}$ from quality-controlled velocities.

| | Tidal slough ADV | Tidal shelf ADV | Under-ice MAVS | Tidal MAVS |
|---|---|---|---|---|
| Measured coordinate system | XYZ | XYZ | ENU | XYZ |
| Rotation method | None | to align with $\bar{U}$ | to align with $\bar{U}$ | None |
| $\delta$ wavenumber range for fitting | 0.8 | 0.8 | 0.8 | 0.8 |
| EPSI in EPSI_FINAL | Vertical (w) | Vertical (w) | Vertical (w) | Transverse (v) |
| Thresholds used when creating EPSI_FLAG | | | | |
| 1. Non-stationary (subsets) | 20 | 20 | 20 | 20 |
| 2. Failed Taylor Hypothesis: $\mathcal{T}_h$ | 0.33 | 0.33 | 0.33 | 0.33 |
| 3. Noise dominated spectra: $\mathcal{T}_n$ | 3 | 3 | 3 | 3 |
| 4. Poor spectral slope: $A$ | 17 | 17 | 17 | 17 |
| 5. Missing velocity samples: $\mathcal{T}_p$ | 10 | 10 | 10 | 10 |
| 6. Anisotropic: $\mathcal{T}_a$ and $L = \kappa L_b$ | 150 and L=0.18 m | 100 and L=0.16 m | 200 and L=2 m | 200 and L=0.58 m |
| 7. Outside inertial subrange | Assumes the inertial subrange ends at $k \approx 0.1/L_k$ | | | |
| 8. User-defined | Not used | Not used | Vortex shedding in u and v | Vortex shedding in u and w |

For our third example, $\hat{\varepsilon}$ estimates from the transverse and vertical components received a boolean $\hat{\varepsilon}$ equal to 32 (Figure 10c). This value translates to applying the 6th flag, concerning turbulence anisotropy ($I \in 6$ in Equation 8). For this dataset, we used a threshold of $\mathcal{T}_A = 100$ to identify segments that were too anisotropic to yield a reliable $\hat{\varepsilon}$ estimate (Equation 24, Table 4). The longitudinal component passed the condition for being sufficiently isotropic, while passing all other quality-control criteria (EPSI_FLAG = 0).

Our fourth and final example received an EPSI_FLAG of 8 in the longitudinal velocity component (Figure 10d). This boolean code implies that this segment failed the 4th criterion, which indicates that the spectral slope $\hat{\beta}_1 = -0.83$ (Figure 11e) was outside the expected range based on Equation 15. For this computation, we used $d = 28.5$ for the spectra's degrees of freedom, $N = 32$ for the number of fitted samples, and $\delta = 0.8$ for the decadal range fitted. The other two velocity components passed all quality-control criteria (EPSI_FLAG=0).





**Figure 10.** Example spectra from four different segments of the *Tidal Shelf ADV* benchmark are shown in separate panels for all three velocity components. The turbulence model spectra for velocities are shown in gray for $10^{-7}$ (darkest) to $10^{-4}\,\mathrm{W\,kg^{-1}}$ (lightest) as digitized by Luznik et al. (2007) from the work of Gargett et al. (1984). The approximate limit between the inertial and viscous subrange for each model spectra is denoted by the diamonds (Equation 4).



Despite not flagging any of the $\hat{\varepsilon}$ from the *Tidal Shelf ADV* for vortex contamination, this contamination source did impact the

reliability of $\hat{\varepsilon}$ of other benchmarks. For example, we flagged all of the MAVS $\hat{\varepsilon}$ estimates obtained from velocity components

perpendicular to the instrument's shaft. This step translates to flagging all the *Tidal MAVS* $\hat{\varepsilon}$ estimates, which were not from

the transverse (y) direction and all *Under-ice MAVS* $\hat{\varepsilon}$ estimates that were not from the vertical (w) direction (Table 4). The

chosen velocity component for the final $\hat{\varepsilon}$ estimates differs between the datasets. The *Tidal MAVS* assigns the $\hat{\varepsilon}$ estimates from

the transverse direction given the orientation of the flow relative to the instrument's shaft, while the other datasets assigned $\hat{\varepsilon}$

from the vertical component. Depending on the intended scientific purposes for the $\hat{\varepsilon}$ estimates, users may want to be more

or less stringent when applying the quality-control metrics. Hence, we recommend documenting the chosen thresholds in the

NetCDF metadata for EPSI_FLAGS.





**Figure 11.** Estimated $\hat{\varepsilon}$ and associated quality control metric used for flagging the *Tidal Shelf ADV* benchmark dataset. a) The 95% bootstrap confidence intervals are shown for the $\hat{\varepsilon}$ estimates that passed all quality-control metrics. b) Taylor Frozen hypothesis (Equation 22) with the mean speed $\overline{U}$ past the sensor on the secondary right axis. c) Noise-dominated spectra metric (Equation 23). d) Too many missing velocity samples comapred to $\mathcal{T}_P$. e) Spectral slopes $\beta_1$ deviate from the expected range (Equation 15). f) Likely anisotropic spectra (Equation 24). g) Boolean flag for our estimated $\hat{\varepsilon}$. The higher EPSI_FLAGS are show in h).



# 6 Conclusions

This paper uses acoustic point-measurement data to describe a systematic approach to obtaining reliable estimates of a key
ocean parameter —the dissipation rate of turbulent kinetic energy $\varepsilon$. We describe the processing and data handling steps, quality
control, and associated flags. Finally, we provide benchmark results for researchers to validate their computer methodologies.

This approach was developed as part of the ATOMIX working group. As such, parallel analyses exist for other ocean
measurement measurement techniques (Lueck et al., 2024). There are benefits to this combined approach, including the ability
to leverage a broader range of experience and coding and making the step from one type of measurement to another much
easier. This benefit also applies to field campaigns with overlapping measurement approaches (e.g., the near-bed section of a
shear profile overlapping with a region measured with a bed-mounted acoustic velocimeter).

One clear but simple conclusion is that there are significant benefits to consistently employing the ATOMIX naming and
storage convention described here. In particular, this enables rapid integration with existing approaches and builds a more
cohesive and efficient sampling community with enhanced cross-talk between researchers using different methods. Over time,
we expect improvements to the best practices as new instruments become available and new environmental conditions are
sampled. With the oceans' continued importance and role in key Earth system processes, more systematic sampling of the
oceans is inevitable. It is important that this sampling produces results that are consistent and reliable.

*Code and data availability.* The benchmarks and tools for loading benchmark datasets are available at the following public repository 10.
5281/zenodo.16798905 (Bluteau et al., 2025) under the SCOR community resources. This repository also includes example templates for
writing our recommended metadata into NetCDF files.

*Author contributions.* Lead author CEB had primary responsibility for writing the manuscript as well as its conception. The other co-authors
contributed to the following sections: Introduction – All co-authors with strong participation from CLS, DW, JM; section 2 and 3 – CLS,
CEB; section 4 – CEB, DW, JM; section 5 – CEB; Conclusion: CLS. All authors contributed to the article and approved the submitted
version. CLS prepared most tables and all schematics. CEB prepared all data-centric figures. All authors critically reviewed the manuscript's
scientific according to the ATOMIX working group's discussion.

*Competing interests.* The authors declare that they have no competing interests.

*Acknowledgements.* These benchmarks were created according to the terms of reference of an international working group #160 ATOMIX,
and the authors thank the other members of the ATOMIX working group (co-chairs Cynthia Bluteau, Ilker Fer and Yueng-Djern Lenn).
The Scientific Committee on Oceanic Research funds this working group through a grant from the National Science Foundation (NSF grant



#OCE-2140395) and contributions from national SCOR committees. CS was supported by Marsden Fund awards NIW1702 and NIW2102. MAVS datasets were provided by Natalie Robinson (supported by the NZ Antarctic Science Platform project ANTA1801) and Alex Hay (*Tidal MAVS*). The ATOMIX wiki has more information about the group's activities, and can be accessed from the working group's website: https://scor-int.org/group/analysing-ocean-turbulence-observations-to-quantify-mixing-atomix/.



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

## Appendix A: NetCDF variable names





**Table A1.** Summary of Level 1 NetCDF data format. This level includes the full resolution raw timeseries velocities in physical units, quality-control flags, and ancillary data.

| Short name<br>Standard name | Dimensions | Comments |
|---|---|---|
| TIME<br>time | TIME | units: Days since a specified reference, e.g., Days since 2005-01-01T00:00:00Z |
| N_VEL_COMPONENT<br>unique_identifier_for_each_velocity_component | N_VEL_COMPONENT | Maximum of 3 for u,v, w (east, north, up) velocities |
| XYZ_VEL<br>water_velocity_measured_in<br>_instrument_coordinates<br>or<br>ENU_VEL water_velocity_measured_in<br>_geographical_coordinates<br>or<br>BEAM_VEL water_velocity_measured_in<br>_beam_coordinates | TIME,<br>N_VEL_COMPONENT | units: m/s<br>reference datum: instrument, geographical, or beam frame of reference for XYZ, ENU or BEAM. The same coordinate system should be used to provide the flags (e.g., XYZ_VEL_FLAGS) and optionally the unwrapped velocities (e.g., XYZ_VEL_UNWRAP_FLAGS) |
| XYZ_VEL_FLAGS<br>water_velocity_measured_in<br>_instrument_coordinates_status_flags | TIME,<br>N_VEL_COMPONENT | CF-compliant 8-bit (0-255) boolean flag that designates why a velocity sample was discarded. |
| HEIGHT or DEPTH | TIME | units: meters |
| *Optional or sensor-dependent* | | |
| BURST_NUMBER<br>unique_identifier_for_each_burst | TIME | Integers of 1, 2, etc, to designate which burst the velocities are associated with. For continuous sampling, this can be omitted or have all samples associated with burst 1. |
| HEADING<br>platform_yaw_angle | TIME | units: degrees<br>positive: clockwise<br>reference datum: true North |
| PITCH<br>platform_pitch_angle | TIME | units: degrees<br>positive: counterclockwise<br>reference datum: around the instrument y-axis |
| ROLL<br>platform_roll_angle | TIME | units: degrees<br>positive: counterclockwise<br>reference datum: around the instrument x-axis |
| ABSIC<br>backscater_intensity | TIME,<br>N_VEL_COMPONENT | units: counts |
| CORRN<br>noise_correlation_percent | TIME,<br>N_VEL_COMPONENT | units: % |
| SNR<br>signal_noise_ratio | TIME,<br>N_VEL_COMPONENT | units: db |
| XYZ_VEL_UNWRAP<br>water_velocity_measured_in_<br>instrument_coordinates_unwrapped | TIME,<br>N_VEL_COMPONENT | units: m s$^{-1}$<br>These velocities needed to be unwrapped owing to choosing an ambiguity velocity too small compared to measured velocities. |



**Table A2.** Summary of Level 2 NetCDF data format. This level includes quality-controlled and segmented timeseries.

| Short name<br>Standard name | Dimensions | Comments |
|---|---|---|
| TIME<br>time | TIME | units: Days since a specified reference, e.g., Days since 2005-01-01T00:00:00Z |
| UVW_VEL<br>water_velocity_in_the_analysis_frame_of_reference | TIME,<br>N_VEL_COMPONENT,<br>N_SAMPLE | units: m/s<br>reference datum: analysis frame of reference<br>Velocities from level 1 stored on a per-segment, i.e., duration $\tau_\varepsilon$ basis. These may be rotated from the original measurement frame stored in level 1. |
| PERGD<br>percentage_of_samples_good | TIME | units: %<br>Percentage of samples in each segment that passed all quality-control metrics. |
| TIME_BNDS<br>time_interval_bounds | TIME, N_BNDS | units: same as TIME<br>Provides the beginning and end of each interval specified by the variable TIME |
| TAYL<br>ratio_of_rms_of_turbulent_velocity_with_mean_water_speed | TIME | Left hand side of Equation 22 |
| ROT_AXIS<br>axis_of_rotation_from_east_to_x | TIME,<br>N_VEL_COMPONENT | units: degree<br>reference datum: east<br>positive: counterclockwise<br>Axis in the geographical coordinate system to rotate velocities into the analysis frame of reference. |
| ROT_ANGLE<br>angle_of_rotation_from_east_to_x | TIME | units: degree<br>reference datum: east<br>positive: counterclockwise<br>Angle for rotating the velocities from geographical coordinates into the analysis frame of reference |
| BURST_NUMBER<br>unique_identifier_for_each_burst | TIME | Integers of 1, 2, etc, to designate which burst the velocities are associated with. For continuous sampling, this can be omitted or have all samples associated with burst 1 |
| N_SAMPLE<br>unique_identifier_for_each_sample_within _the_segment | N_SAMPLE | Value from 1 to $f_s\tau_\varepsilon$ to designate the velocity sample in each segment, and thus the largest value is based on sampling frequency $f_s$ and segment duration $\tau_\varepsilon$ |
| N_BNDS<br>unique_identifier_for_defining_low_high_bounds | 1,2 | 1 represents the lower bound and 2 the upper bound |
| N_VEL_COMPONENT<br>unique_identifier_for_each_velocity_component | N_VEL_COMPONENT | Same as level 1 |





**Table A3.** Summary of Level 3 NetCDF data format. This level includes the spectral observations.

| Short name<br>Standard name | Dimensions | Comments |
|---|---|---|
| TIME | TIME | Same as level 2 |
| TIME_BNDS | TIME, N_BNDS | Same as level 2 |
| UVW_VEL_SPEC<br>power_spectrum_density_of_velocity_in_the_analysis_<br>frame_of_reference | FREQ,<br>N_VEL_COMPONENT,<br>TIME | units: $(\text{m s}^{-1})^2$/Hz<br>reference datum: analysis frame of reference<br>Summing these spectra across all frequencies should equal the signal's variance estimated in the time-domain. |
| FREQ<br>frequency | FREQ | units: Hz |
| PSPD_REL<br>platform_speed_wrt_sea_water | TIME | units: $\text{m s}^{-1}$<br>Mean speed past the sensor $\bar{U}$ used to convert from frequency (time) to wavenumber (space). |
| DOF<br>degrees_of_freedom_of_spectrum | 1 | See Equation 11 since it depends on how the spectra was computed. |
| SPEC_NOISE_UVW<br>power_spectrum_density_white_noise_of_velocity_in_the_<br>analysis_frame_of_reference | TIME,<br>N_VEL_COMPONENT | units: $(\text{m s}^{-1})^2$/Hz<br>Typically determined from the high-frequency (noise-dominated) part of the spectrum i.e., noise floor (§4.4.3). |
| KVISC<br>kinematic_viscosity_of_water | 1 or TIME | units: $\text{m}^2 \text{ s}^{-1}$ |
| BURST_NUMBER<br>unique_identifier_for_each_burst | TIME | Same as level 2 |
| N_VEL_COMPONENT<br>unique_identifier_for_each_velocity_component | N_VEL_COMPONENT | Same as level 1 |
| N_BNDS<br>unique_identifier_for_defining_low_high_bounds | 1,2 | Same as level 2 |





**Table A4.** Summary of Level 4 NetCDF data format. This level includes timeseries of the $\hat{\varepsilon}$ dissipation estimates. The parameters necessary for re-flagging $\hat{\varepsilon}$ estimates are shown separately in Table A5.

| Short name<br>Standard name | Dimensions | Comments |
|---|---|---|
| EPSI<br>specific_turbulent_kinetic_energy_dissipation_in_water | TIME,<br>N_VEL_COMPONENT | units: W kg$^{-1}$<br>$\hat{\varepsilon}$ estimated from each of the individual velocity component. |
| EPSI_FLAGS<br>specific_turbulent_kinetic_energy_dissipation_in_water<br>status_flag | TIME,<br>N_VEL_COMPONENT | units: W kg$^{-1}$<br>See Table 3. CF-compliant 8-bit (0-255) boolean flag that designates why an $\hat{\varepsilon}$ estimate was flagged as being of poor quality. |
| EPSI_CI<br>specific_turbulent_kinetic_energy_dissipation_in_water confidence_interval | TIME,<br>N_VEL_COMPONENT | units: W kg$^{-1}$<br>95% confidence interval from bootstrapping residuals |
| EPSI_FINAL<br>specific_turbulent_kinetic_energy_dissipation_in_water_final | TIME | units: W kg$^{-1}$<br>comment: Specifies which velocity component was retained as the final $\hat{\varepsilon}$ estimates that would be provided in a scientific publication. |
| EPSI_FINAL_CI<br>specific_turbulent_kinetic_energy_dissipation_in_water<br>final_confidence_interval | TIME | units: W kg$^{-1}$<br>95% confidence interval of the final $\hat{\varepsilon}$ from bootstrapping residuals. |
| TIME | TIME | Same as level 2 |
| TIME_BNDS | TIME, N_BNDS | Same as level 2 |
| PSPD_REL | TIME | Same as level 3 |
| K_BNDS<br>fitted_wavenumber_bounds_of_spectra | TIME,<br>N_VEL_COMPONENT,<br>N_BNDS | units: (m s$^{-1}$)$^2$ per cpm<br>Provides the first and last wavenumber bound fitted to estimate $\hat{\varepsilon}$ |
| BURST_NUMBER | TIME | Same as level 2 |
| N_VEL_COMPONENT | N_VEL_COMPONENT | Same as level 1 |
| N_BNDS | 1,2 | Same as level 2 |



**Table A5.** Additional parameters stored at Level 4 NetCDF for re-flagging $\hat{\varepsilon}$ estimates.

| Short name Standard name | Dimensions | Comments |
|---|---|---|
| PERGD | TIME | Same as level 2 |
| TAYL | TIME | Same as level 2 |
| MIN_EPSI_NOISE minimum_specific_turbulent_kinetic_energy_dissipation in_water_resolvable | TIME, N_VEL_COMPONENT | units: W kg$^{-1}$ Calculates $\varepsilon_m$ using Equation 18 with the highest fitted wavenumber $k$. |
| L turbulent_length_scale | 1 or TIME | units: m comment: Should specify the definition used for estimating the largest turbulent overturn, which could be $L_O$, $L_S$, or $L_z = \kappa z$ (Equation 5, 6, 7). |
| KVISC | 1 or TIME | Same as level 3 |
| SPEC_SLOPE estimated_spectral_slope_of_fitted_wavenumbers _in_logspace | TIME, N_VEL_COMPONENT | $\beta_1$ in $\hat{\Psi} \sim k^{\beta_1}$ (Equation 2) |
| DECADE | 1 | The fitted wavenumber range $\delta$ given by Equation 14 and needed for calculating the acceptable $\hat{\beta}_1$ range in Equation 15 |
| N_FITTED number_of_fitted_samples | TIME, N_VEL_COMPONENT | $N_s$ in Equation 15 |
| DOF | 1 | Same as level 3. Required for calculating the acceptable slope $\hat{\beta}_1$ range in Equation 15 |
| DIR_CSPD direction_of_water_speed | TIME | units: degree reference datum: x-axis positive: counterclockwise Useful for flow interference. |