# Peer review of "Best practices for estimating turbulent dissipation from oceanic single-point velocity timeseries observations"

_EGUsphere, 2025_

## Author Comment (AC1)

**REVIEWER 1**

*This submission provides guidance and recommendations for the observation and estimate of turbulent dissipation from single-point velocity time series that were developed by the SCOR Working Group #160 to quantify ocean mixing. It is exclusively a methods-type manuscript and does not report new science or observations, and is founded upon previous publications by the lead author such as Bluteau et al. (2011) with which there is some overlap. This contribution provides useful guidance, but there are a number of mostly minor clarifications that should be addressed before publication.*

We thank Reviewer 1 for their feedback and respond to their comments below.

**Main comments**

*Figure 2: the caption should be revised to explain the multiple frequency axes above the figure that correspond to different mean advection speed. Also, there are inconsistencies between terminology used for units in this figure: m s $^{-1}$ in the upper axes, m/s in the y-axis.*

We now explain the different frequency axes in the caption, and updated the y-axis label.

*Figure 3: I'd suggest updating the caption for this figure to improve clarity. Describe the pathway around the figure more explicitly perhaps with the aid of a few more labels. Also, cross-reference it more thoroughly with the main text when the stages are being discussed.*

The figure was simplified to highlight the four panels represent different processing levels, each stored in their own group within the NetCDF file. The caption was also modified to describe the data processing associated with each level.

*Figure 4: in the caption, please can you explain why a power of 0.2 was applied to the number of samples n in each histogram bin.*

This was done simply to reduce the range so that location of n=0 is more clearly visible in the histogram; it has no bearing on the results. This follows Brock, which uses $n^{0.5}$. A comment to this effect has been added to the caption.

*Figure 5: could I suggest increasing the line thickness of the length scales and durations to aid cross-correlating them with the colorbars.*

We increased the line thickness in the two contour plots.

*Line 355: logLAD undefined*

We have amended the text on L394 to state the method involves minimizing the absolute residuals and referred the reader to the numbered equation in *Bluteau* (2025) since there's an entire section dedicated to this method and all other methods assessed for spectral fitting in that paper.

> *Line 561: typo "...velocities samples..."*

This was changed to "velocity samples" on L622.

> *Figure 11: please clarify the colouring and shading of the markers in panel (a). The legend for flagged data in panel (b) overlaps the plotted data and is confusing.*

The caption now states that "If no error bars are presented, then the estimate was flagged for not meeting one or several quality-control criteria." The legend in panel b was moved and is now boxed to avoid confusing it with a data point.

> *It is very helpful that the authors provide sample datasets via the supplementary repository, but have they considered providing some sample Matlab or Python code to perform some of the more standardised aspects of the data preparation and analysis process?*

Prior to writing this manuscript, the development of the Working Group's Terms of Reference examined the possibility of including code. Ultimately it was decided against distributing software (see Terms of Reference). The logic for this decision was the overarching objective being to document the best practices so that researchers can then validate different processing workflows against the benchmarks at any checkpoint (e.g., fitting algorithms at Level 3). Our chosen strategy avoids the pitfall of having the working group's activities become obsolete because the code cannot be maintained and updated to keep up with new flavour of programming languages. It also intrinsically recognises that there may be other ways to arrive at the same answer.

**REFERENCES**

Bluteau, C. (2025), Assessing statistical fitting methods used for estimating turbulence parameters, *Limnol. Oceanogr.: Methods*, doi:10.1002/lom3.10729.

---

## Author Comment (AC2)

REVIEWER 2

*Estimates of TKE dissipation rate from inertial subrange fitting have not necessarily used consistent processing procedures from different groups, etc. The authors seek to publish a set of best practices for future research that follows. Although it does not include new scientific findings, I think this would be within the scope of EGU Ocean Science as a Technical Note and has potential to be useful to the community. I do have some minor comments and suggestions that I think would improve the paper.*

*1.The main point of this manuscript is to establish a set of "best practices". While I agree that the methods in the manuscript are thorough and robust, I think it might be useful to discuss previous research in more detail and use this to guide justification of the new methods; i.e., how do the proposed methods differ from what has been used in the past and why are they superior? This, I think, would be important for 1) convincing readers that the proposed methods are indeed the best practice and 2) ultimately, having the community adopt them. This is probably best addressed in several different places throughout the text rather than in one place. To be fair, the authors have clearly thought about this issue and there are components of it already in the text. For example the justification on L218-224 is well referenced and is sort on in-line with what I think would improve the overall impact of the paper. However, I do not think the justification in other places is quite as thorough and convincing. A few places of these are pointed out below.*

*2. I suggest that the authors consider sharing relevant parts of the code used to prepare and process the sample datasets. I think the main intention of the authors is for their methods to ultimately be utilized by the community. While my suggestion is optional, sharing code would make it much easier for others to adopt the suggested practices.*

We thank Reviewer 2 for their feedback and respond to their comments below. Notably, we have expanded more on the choices of methods whenever feasible e.g., linear interpolation of quality-controlled data [L307-314]. However, we have not provided code. Prior to writing this manuscript, the development of the Working Group's Terms of Reference examined the possibility of including code. Ultimately it was decided against distributing software (see Terms of Reference). The logic for this decision was the overarching objective being to document the best practices so that researchers can then validate different processing workflows against the benchmarks at any checkpoint (e.g., fitting algorithms at Level 3). Our chosen strategy avoids the pitfall of having the working group's activities become obsolete because the code cannot be maintained and updated to keep up with new flavour of programming languages. It also intrinsically recognises that there may be other ways to arrive at the same answer.

**Line-by-line comments and suggestions:**

*L32 – Include the full name of MAVS when first stated. I see it later on, but it should be included here*

Done on L34.

*L30-36 – I suggest to elaborate more on the reasons why the approaches are differently suited to high and low energy regimes.*

We have expanded more on L25-30 about the differences between shear probes and velocity sensors. Namely, shear proves can resolve the viscous subrange, and thus can sample in lower energy environments than velocity sensors.

> *Fig 2 – I suggest to remove the gray shading inside the triangles. It suggest that the shading represents dissipation rate, but that is not the case.*

Done.

> *L81-82 – Add additional clarification. That is, I think the authors are referring to the longer-term distribution of epsilon here, not that epsilon itself does not change over faster timescales than the sampling frequency (which as the authors are surely aware can happen and does not necessarily signify a violation of stationarity).*

> *Reading on, I see the authors include a nice explanation of this at L259... maybe also briefly mention it here.*

As suggested, the text on L93-95 was modified by using elements of the explanation that had been given later on.

> *L83-84 – Perhaps move this to the start of the paragraph?*

Done.

> *L121 – I suggest to add a citation or additional explanation*

We added a reference to *Bluteau et al.* (2011) given their discussion on the use of the distance to the boundary (eq. 7) in lieu Corrsin length scale (eq. 6).

> *L127-137 – Perhaps Fig 2 should be referred to at the start of the paragraph? It visually explains much of what is here, and I think may be easier to understand for readers new to the topic*

Figure 2 is now referenced at the beginning of this paragraph at L140.

> *Table 1 title – U bar is 5th and 95th percentile, correct? Also, why not 2.5 and 97.5 to get the 95% confidence interval.*

No it was 50th (median) and 95th percentile. We now provide the 2.5th, 50th, and 97.5th percentile despite the 2.5th percentile being near zero when the tide turns.

*L154-159 – Although important for understanding the datasets, I feel this is a bit of an aside from the main text. If this is a suggestion as the format for data storage for future experiments, that should be clearly stated. From my experience, many (most?) published datasets are essentially just the "fourth level", and if there is a suggestion for the inclusion of more information that should be discussed. In any case, I think the main objective is the methods for how to get epsilon, and perhaps this section can be pared down.*

*Reading on, this is necessary to explain for section 4, so maybe it can be incorporated into that section*

We have revised and shortened this paragraph and the beginning of this section to emphasize the purpose of the benchmarks and why we have four groups. We now also explicitly state which data should be submitted with peer-reviewed manuscripts and what should be summarised in the publications' methods section [L663-667].

*L162 – I think linear interpolation needs to be justified here. I understand the necessity to fill in gaps for constructing spectra, and think further explanation would be useful.*

This section is just an overview of the steps, so we have removed the reference to linear interpolation; options for replacing missing data are now discussed more in Section 4.2.

*L164 – I don't understand what "turbulence models" means in this context. I don't think it's the method of calculating epsilon, since that would still be the same within a dataset?*

It referred to the choice of turbulence equation (model) to fit the spectra. We have revisited the sentence accordingly on L175.

*Sec 4.1 (until L211) – I'm not sure how useful parts of this section are. That is, I don't know if it is useful to state that QC should be done on a case-by-case basis or to follow manufacturer recommendations. I understand it is important to QC and thus why the authors have discussed this, but perhaps it could be shortened in places.*

The introduction to the processing method was shortened and re-organised so that basic QAQC and phase-wrapping appear in their own subsections before discussing despiking.

*Fig 4 – I suggest to put a box around the legend in panel a) for clarity and consistency with panel b).*

The legend box is now added to panel a).

*L312-321 – See my general comment. This is a place where I think additional background information, and references that highlight deficiencies of other methods, would be useful. E.g. why is this spectral averaging procedure different and superior to others?*

Some elements of spectral analysis cross over between ATOMIX groups. The ATOMIX paper on shear probes discuss this particular point (*Lueck et al.*, 2024) so we have added a clearer pointer to that here.

> *L340-356 – I think additional background information and referencing would be useful here, as well. Is there another study that has found this with observations, in addition to the study using synthetic spectra which is already cited?*

There aren't other papers with real observations. Besides the true $\varepsilon$ is unknown with real observations making the assessment of fitting methods challenging. We have added a paragraph at L354-363 summarizing the six methods tested in the cited paper by *Bluteau* (2025a), which was written solely to address the lack of papers on testing of fitting methods.

> *L346,355,357 – "loglad" and "ladlog" are the same thing, right? Please clarify. These are not clearly defined.*

Yes, this is a typo. They should all be labelled logLAD.

> *Fig 6 – I don't understand the purpose of showing Fig 6. Is it just to show the values of "A"? These are already stated in the text, and I barely see any difference between the different colored lines (I think that is to show insensitivity to the sampling, perhaps?). Maybe I am missing something here.*

Figure 6 (now 8) is meant to visually illustrate that the functional form of the expected bounds for the calculated spectral slopes in equation 24. The figure has been moved much later in section 4.6.4 when discussing flagging $\hat{\varepsilon}$.

We modified the y-label to highlight that it represents the $A$ thresholds that must be selected by users to flag their data. We have added the individual data points for each performed tests, which now highlights the maximum deviation from the known (prescribed) slope $\beta_1 = -5/3$ occurs when $A = 17$.

Yes, the lines being flat means $A$ remains consistent across the four series of numerical experiments conducted across a wide range degrees of freedom $d$ (x-axis). Each series were conducted against 3200 synthetic spectra (100 per degrees of freedom tested) (*Bluteau*, 2025a,b).

> *Sec 4.4.2 – Similar to my previous comments, are there any other studies that have tried different inertial ranges, especially with field observations? Or that have found deficiencies with a method or specifications that are not recommended in the present manuscript? While I agree the results of Bluteau 2025 for synthetic observations cited here and elsewhere are very relevant, I think additional background would strengthen some of the recommendations.*

We didn't find any literature that tested methods for finding the inertial subrange. Many studies fix the wavenumbers being fitted, and presume it won't include the low and high wavenumbers flattened by the impact of the mean flow (energy containing range) and noise, respectively.

Below, we illustrate the differences among the methods against our recommended "slope" method for synthetic spectra (Figure r1) and the *Tidal Shelf ADV* spectra (Figure r4 and r5). The tests against the synthetic

spectra show that the three other methods tend to situate the inertial subrange at low wavenumbers (Figure r2 a-c), albeit at higher wavenumbers than the slope method (Figure r2 d-f). Nevertheless, the returned $\hat{\epsilon}$ generally agreed with the prescribed value $\epsilon_p$ (Figure r2 d-f) since the inertial subrange is sufficiently broad and all methods correctly avoided the noise-dominated portion of the spectra when searching for the inertial subrange.

When noise levels were increased by an order of magnitude to $\Phi_n = 10^{-6}$ (m/s)$^2$/rad/m, the identified wavenumbers tended to sit at low wavenumbers for all four methods since the noise floor now impinges on the high wavenumbers of inertial subrange (Figure r3 a-c). There was still a tendency for our recommended method to situate the inertial subrange at lower wavenumbers than the other three methods (Figure r3 d-f). The estimated $\hat{\epsilon}$ tends to deviate more when the identified wavenumbers from the other methods deviate from those determined by our recommended "slope" method (Figure r3 d-f). However, these deviations tend to be associated with spectra that have low degrees of freedom i.e., with minimal spectral averaging (not shown).

We also did some tests where both the low wavenumbers were flattened (not shown) to replicate another common problem with observed spectra deviating from the expected model form. The estimated $\hat{\epsilon}$ agreed better with the prescribed value when the slope method was used to locate the inertial subrange, and this despite situating it at lower wavenumbers than the three other methods.

For illustration purposes, we also applied the four identification methods against our *Tidal Shelf ADV* benchmark dataset. The comparison between our recommended "slope" method with the median of $|\log\hat{\Psi} - \log\Psi|$ is presented in Figure r4. The larger discrepancies tend to be associated with very poor spectra (Figure r5a) that end up being flagged later on for other reasons besides the poor slope.

We have added a bit more information about the testing in the manuscript in §4.4.2, although we do note that all methods for identifying the inertial subrange do fairly well.

> *Also, I'm curious if there was a distinction or impact on performance when only short segments had a good-enough fit to be classified in the inertial range, as opposed to cases where longer segments fit $k^{-5/3}$.*

For completeness, we present equivalent plots for the tests done over much shorter subsets ($\delta = 0.3$, Figure r6 and r7) than those described above ($\delta = 1$, Figure r2 and r3). Compared to the tests with longer subsets ($\delta = 1$), the identified wavenumbers tend to become more variable for each method, especially for the spectra with lower noise levels (Figure r6 a-c vs r7 a-c). The estimated $\hat{\epsilon}$ also deviated more from the prescribed $\epsilon_p$ across the three "non-slope" methods when the noise floor increased from $\Phi_n = 10^{-7}$ to $\Phi_n = 10^{-6}$ (m/s)$^2$/rad/m (Figure r6 d-f and r7 d-f), albeit the deviations were less pronounced than for the tests with longer subsets (Figure r2 and r3).

Regardless of the performance difference when using short subsets, we recommend using fairly long subsets $\delta$ to identify the location of the inertial subrange so that it matches the recommended $\delta \geq 0.8$ noted at L424 for estimating $\hat{\epsilon}$ from fitting velocity spectra.

> *L504 – A=17 is dependent on the estimates you made, and not universal, correct? That is implied at L507, I think. I suggest to reword here (and in the other subsections of 4.6 where applicable) whether the suggested thresholds are expected to be reasonable in all cases.*

Most of the text surrounding $A$ has now been moved into section 4.6.4 when discussing the flagging of $\hat{\epsilon}$ estimates. $A$ is user-defined and has been empirically determined by scaling the results in Figure 7 and 8 of

*Bluteau* (2025a) with the main parameters responsible for the variations in estimated $\hat{\beta}_1$ across their series of numerical experiments. The user-selected $A$ presumes that the functional form of the expected bounds for the calculated spectral slopes in equation 24 is correct. The estimated $A$ for each individual spectra tested have been added in Figure 8 to highlight the maximum $A = 17$ in addition to 99.7% bounds that were already illustrated.

> *Sec 4.6.6 – Wouldn't anisotropy potentially also influence the spectral shape? Also, related to this, it might be worth mentioning that these flags are not exclusive at the start of section 4.6.*

Yes, it does impact the spectral shape which is why we have section 4.4.2 and Figure 6 to discuss how to identify the inertial subrange. This section emphasizes the need for identifying the inertial subrange to avoid the flatter spectra at low wavenumbers (anisotropy) and high wavenumbers (noise).

The text in §4.6.6 was also revised since this flag is mostly for identifying segments when the turbulence levels are too weak to be considered isotropic as opposed to flagging spectra with flatter shapes at low wavenumbers.

> *L563 – I don't quite understand why this flag was not applied. Is it because phase unwrapping was easily rectified (as I think the earlier text might suggest)?*

The flag was applied and is stored in the NetCDF. The sentence *"For this dataset, the 8th flag, denoting phase-wrapped samples, was not used to discard velocity samples"* was rewritten since the velocity samples were unwrapped for subsequent analysis rather than discarded, unlike samples flagged for any other reason.

> *Fig 11e – Like Fig 4, I suggest to put a box around the legend as the legend symbol for "flagged" is hard to distinguish from the actual flagged values. This could also be done for other panels in this figure.*

Each legend in Fig 11 is now boxed.

**Figures about identifying inertial subrange**

[Figure]

**Fig r1:** Example spectra used to test the four methods described in §4.4.2 of the manuscript for identifying the inertial subrange. These tests' results are shown in Fig r2, r3, r6, and r7. These tests were against the 3200 synthetic spectra with their uncertainty generated using the $\chi_d$-distribution (*Bluteau*, 2025b,a). White noise at a pre-specified energy level was added to each of the 3200 synthetic spectra to mimic noise floor impinging high wavenumbers. Two examples are shown for two different degrees of freedom $d$ in with a (a) noise floor of $\Phi_n = 10^{-7}$ (m/s)$^2$/rad/m and (b) with a higher noise floor of $\Phi_n = 10^{-6}$ (m/s)$^2$/rad/m. The prescribed $\varepsilon_p = 10^{-6}$ W kg$^{-1}$ and the inertial subrange ends approximately at $\hat{k} \approx$ 13cpm.

REFERENCES

Bluteau, C. (2025a), Assessing statistical fitting methods used for estimating turbulence parameters, *Limnol. Oceanogr.: Methods*, doi:10.1002/lom3.10729.

Bluteau, C. (2025b), Synthetic spectra for assessing statistical fitting methods used to estimate ocean turbulence, *Zenodo*, doi:10.5281/zenodo.10576543, zenodo.

Bluteau, C. E., N. L. Jones, and G. N. Ivey (2011), Estimating turbulent kinetic energy dissipation using the inertial subrange method in environmental flows, *Limnol. and Oceanogr.: Methods*, *9*, 302–321, doi: 10:4319/lom.2011.9.302.

[Figure]

**Fig r2:** Left panels: comparison of the median wavenumber identified as being the most likely location for the inertial against those determined from our recommended "best slope" method for the three other methods tested: mad (a), mean of $|\log \hat{\Psi} - \log \Psi|$ (b), median of $|\log \hat{\Psi} - \log \Psi|$ (c). The right panels (d) to (f) show the deviations of the estimated $\hat{\varepsilon}/\varepsilon_p$ for the three methods presented in y-axis of the left panels (a)-(c), respectively. The x-axis represents the ratio of the results presented in the left panels. The fitted decadal range for each subset was $\delta = 1$ while the noise floor added was $\Phi_n = 10^{-7}$ (m/s)$^2$/rad/m (see example synthetic spectra in Figure r1 a).

[Figure]

**Fig r3:** Same as Figure r2 but adding a noise floor of $\Phi_n = 10^{-6}$ (m/s)$^2$/rad/m (see example synthetic spectra in Figure r1 b).

[Figure]

**Fig r4:** The returned $\varepsilon$ pending which method is used to determining the location of the inertial subrange. Using the minimum from the median of $\Delta \log \Psi = |\log \hat{\Psi} - \log \Psi|$ returns generally similar results as the $\varepsilon$ corresponding to the spectra with the slope closest to -5/3. The observations are from the *Tidal Shelf ADV* benchmark.

Lueck, R., et al. (2024), Best practices recommendations for estimating dissipation rates from shear probes, *Frontiers in Marine Science*, *11*, doi:10.3389/fmars.2024.1334327.

[Figure]

**Fig r5:** Similar to Figure 6 in the manuscript. This segment had the biggest discrepancy in Figure r4 for where the inertial subrange was placed in the spectra (segment 13 from Tidal Shelf ADV experiment).

[Figure]

**Fig r6:** Left panels: comparison of the median wavenumber identified as being the most likely location for the inertial against those determined from our recommended "best slope" method for the three other methods tested: mad (a), mean of $|\log\hat{\Psi} - \log\Psi|$ (b), median of $|\log\hat{\Psi} - \log\Psi|$ (c). The right panels (d) to (f) show the deviations of the estimated $\hat{\varepsilon}/\varepsilon_p$ for the three methods presented in y-axis of the left panels (a)-(c), respectively. The x-axis represents the ratio of the results presented in the left panels. The fitted decadal range for each subset was $\delta = 0.3$ while the noise floor added was $\Phi_n = 10^{-7}$ (m/s)$^2$/rad/m (see example synthetic spectra in Figure r1 a).

[Figure]

**Fig r7:** Same as Figure r6 but adding a noise floor of $\Phi_n = 10^{-6}$ (m/s)$^2$/rad/m (see example synthetic spectra in Figure r1 b).

---

## Author Comment (AC3)

**REVIEWER 3**

*This paper provides a standardized method of processing measurements of epsilon from point velocimeters. It provides clearcut recommendations for all aspects of processing, from QC and processing of velocity data through the fitting of the inertial subrange spectra to retrieve epsilon. This paper, and papers like it, provide a valuable opportunity to make turbulence measurements more accessible to the oceanographic community.*

We thank Reviewer 3 for their feedback and respond to their comments below.

**General comments**

*To echo other reviewers, I think code examples of your processing methodology would be extremely useful in allowing readers to actually implement your suggested best practices. Please consider it. A similar field turbulence methods paper (Zippel et al; doi: 10.1175/JTECH-D-21-0005.1) provided their code. It doesn't need to be perfectly commented or organized!*

Prior to writing this manuscript, the development of the Working Group's Terms of Reference examined the possibility of including code. Ultimately it was decided against distributing software (see Terms of Reference). The logic for this decision was the overarching objective being to document the best practices so that researchers can then validate different processing workflows against the benchmarks at any checkpoint (e.g., fitting algorithms at Level 3). Our chosen strategy avoids the pitfall of having the working group's activities become obsolete because the code cannot be maintained and updated to keep up with new flavour of programming languages. It also intrinsically recognises that there may be other ways to arrive at the same answer.

*There are many different ways to approach parts of the analysis such as spectral fitting and selection of the wavenumbers that define the inertial subrange. You make a case for your chosen method, but I don't think that enough evidence is always given to justify your choices as "best practices". Please see specific instances in the line-by-line comments*

There are indeed processing steps for which the results were insensitive to the choice of techniques/method. In those situations, we elected to recommend the easiest to implement by users. We have also added some introductory text to emphasize that this best practices represents a starting point towards developing standards as the community implements and develops new techniques. The data format was specifically designed to facilitate testing at intermediary points of the processing workflow and build upon the working group's activities [L54-58].

**Line-by-line comments**

*Line 26: Is it possible to briefly explain the physical reason that microstructure profilers are better suited for low energy environments and point-velocity better for high-energy?*

We have added more text to highlight that the difference is attributed to shear probes being able to resolve the viscous (smallest) scale of turbulence, while the point-velocity sensors can typically only resolve the larger scales of turbulence. The viscous subrange are the last to be negatively impacted by flow properties (e.g., anisotropy), and thus lowers the resolvable $\varepsilon$ that can be estimated.

*Line 57: Since Shcherbina et al, 2018, at least a couple of papers have shown pulse-coherent ADCPs to be a viable method for obtaining field measurements of dissipation rate, so perhaps the phrasing "on the cusp" is inaccurate. Please see Zippel et al. (2021); "Moored Turbulence Measurements Using Pulse-Coherent Doppler Sonar"; doi: 10.1175/JTECH-D-21-0005.1*

The Zippel et al. citation was added. The text was also reworded to highlight that smaller bin sizes of ADCPs now allow for direct computation of the wavenumber spectra.

*Line 70 u_rms should be defined*

This term has been defined on L77.

*Line 76: Please define and explain "ambiguity velocity".*

As suggested we have clarified "The ambiguity velocity defines the maximum (unambiguous) along-beam velocity that can be measured (for a given transmit pulse)".

*Line 84: Because you go on to also describe the viscous subrange and "large turbulent scales", perhaps a description of the turbulence cascade would be helpful here.*

We added some text to situate the inertial subrange within the other subranges on L93-95 by briefly stating the energy cascade model.

*Line 94: "The largest scales [of the inertial subrange]" – please specify in order to avoid confusion with the largest scales of turbulence in general.*

The sentence was reworded as "being more comparable" to the smallest scales. The largest scale of the inertial subrange according to the cited references in that sentence are L/3, thus about a third of the largest scales of turbulence "in general".

*Line 127: Do you mean "highest" rather than "high"?*

No, high not the highest. A prime example is the Tidal_MAVS dataset in Figure 1. We are not even close in resolving the highest wavenumbers within the inertial subrange, but because it's an energetic site, we resolve ample of the inertial subrange to estimate $\hat{\varepsilon}$.

*Line 203: I think it would be worth going into more detail on how unwrapping can be performed, or at least providing a reference to a paper that does. Some wrapping is unavoidable, and even datasets with a lot of wrapping can still be fully usable once corrected.*

As suggested, we have added some relatively recent references to recent works that outline a number of methods to undertake phase-unwrapping.

*Line 204 More detail on selecting the max velocity during programming in order to avoid wrapping should be given. For example, if you've never sampled in a particular environment before, how should you best estimate the velocity range? Perhaps you could refer to the velocity ranges given in Table 1 as examples of what a user could expect across different environments.*

The phase wrapping is now described in its own section 4.1.1. We have expanded on ways to choose the maximum velocities (numerical modelling, tidal and hydrological predictions). We also highlighted that sometimes you still get it wrong despite having much of the required information in hand (e.g., phase-wrapped *Tidal Shelf ADV benchmark*).

*Line 330: Even though these methods are detailed in Bluteau (2025), it would be good to at least summarize what they are here.*

Yes, we have added a paragraph at L356-364 summarizing the six methods tested by *Bluteau* (2025)

*Line 333: Why was this method chosen of the 6? Perhaps more details should be given on the comparison analysis between the 6 stated methods?*

We added several sentences about the better accuracy of log-based methods for spectra with low degrees of freedom (i.e., computed from short time segments), and explained the advantage of our recommended logLAD method over the least-squares of log-transformed spectra [L368-371].

*Line 365: How does a user decide on a value for A? Perhaps mention that you describe this further in Section 4.6.4. Also, why not base flagging directly on the deviation of the spectral slope from the theoretical -5/3? (I am not suggesting that is better, I'm just having some trouble following how A is derived)*

We refer the reader now to section 4.6.4 where the choice of A is discussed when flagging. We do not recommend flagging the deviation of the estimated slope from -5/3 because the estimated slope varies (within known bounds shown later) depending on the technique used and the amount of spectra averaging when estimating the spectra (hence *A*).

*Line 375: Here, does log-transformed spectra refer to the (synthetic) observed spectra ($\hat{\Psi}$), and model to the theoretical spectra with the $-5/3$ slope (Psi)?*

Yes, the observed (or synthetic) spectra depending on the context. The sentence has been amended with the addition of equation 16. The identified inertial subrange from these two methods are now illustrated in Figure 6 d,e.

*Line 376: Why is this particular strategy recommended?*

We have provided more information in §4.4.2 as to why we recommended this strategy. We chose the slope method partly for convenience since we are calculating the slope ($\beta_1$) to flag $\hat{\varepsilon}$. This method was less biased than the other methods when fitting spectra with limited degrees of freedom provided the fitting uses our recommended decadal range $\delta \geq 0.8$ (not shown). However, for spectra that are sufficiently averaged (i.e., degrees of freedom $d \geq 10$), all four methods returned acceptable results for locating the inertial subrange.

*Line 399: Am I understanding that Figure 8 is showing that the introduction of noise does not cause substantial deviations in the ratio of theoretical to measured epsilon? I'm not sure that this alone is enough to justify your method of treating noise is optimal.*

Figure 8 (now Figure 7) shows by how much $\varepsilon$ can be over-estimated based on which portion of the inertial subrange is fitted and spectra's noise levels. It is used to justify flagging $\hat{\varepsilon}$ based on an acceptable deviations from $\varepsilon$ based on the fitted wavenumbers and noise levels. We have added a few sentences to explain why it was chosen over the other strategies found in the literature [L437-440].

*Figure 8: Please define k, L_K, epsilon, and epsilon_m in the caption. Interpretation of figures is greatly helped by avoiding the reader having to hunt through the text for variable definitions.*

We added the word minimum in front of $\varepsilon_m$, and written in full k and $L_k$.

**REFERENCES**

Bluteau, C. (2025), Assessing statistical fitting methods used for estimating turbulence parameters, *Limnol. Oceanogr.: Methods*, doi:10.1002/lom3.10729.